# Habitat and social factors shape individual decisions and emergent group structure during baboon collective movement

Ariana Strandburg-Peshkin[1]*[†], Damien R Farine[2,3,4], Margaret C Crofoot[5,6,7], Iain D Couzin[2,3]

[1]Department of Ecology and Evolutionary Biology, Princeton University, Princeton, United States; [2]Department of Collective Behaviour, Max Planck Institute for Ornithology, Konstanz, Germany; [3]Department of Biology, Chair of Biodiversity and Collective Behaviour, University of Konstanz, Konstanz, Germany; [4]Department of Zoology, Edward Grey Institute of Field Ornithology, University of Oxford, Oxford, United Kingdom; [5]Department of Anthropology, University of California, Davis, Davis, United States; [6]Animal Behaviour Graduate Group, University of California, Davis, Davis, United States; [7]Smithsonian Tropical Research Institute, Panama

*For correspondence: arianasp@gmail.com

Present address: [†]Department of Migration and Immuno-Ecology, Max Planck Institute for Ornithology, Radolfzell, Germany

**Abstract** For group-living animals traveling through heterogeneous landscapes, collective movement can be influenced by both habitat structure and social interactions. Yet research in collective behavior has largely neglected habitat influences on movement. Here we integrate simultaneous, high-resolution, tracking of wild baboons within a troop with a 3-dimensional reconstruction of their habitat to identify key drivers of baboon movement. A previously unexplored social influence – baboons' preference for locations that other troop members have recently traversed – is the most important predictor of individual movement decisions. Habitat is shown to influence movement over multiple spatial scales, from long-range attraction and repulsion from the troop's sleeping site, to relatively local influences including road-following and a short-range avoidance of dense vegetation. Scaling to the collective level reveals a clear association between habitat features and the emergent structure of the group, highlighting the importance of habitat heterogeneity in shaping group coordination.

## Introduction

Across a wide range of taxa and habitats, animals exhibit coordinated group movement. Maintaining cohesion with conspecifics can benefit individuals by reducing predation risk, improving foraging efficiency (*Krause and Ruxton, 2002*), and, in some cases, by enhancing sensing of environmental gradients (*Berdahl et al., 2013*). For social animals moving in complex, heterogeneous environments, individual decisions and the resulting collective movement patterns are likely to be driven by a combination of factors, including habitat and social influence. Despite this, studies of collective motion have tended to discount the potential role of environmental complexity by either conducting experiments in relatively simple or featureless environments composed of mostly unobstructed space, as in many lab experiments (*Buhl et al., 2006*; *Katz et al., 2011*; *Herbert-Read et al., 2011*; *Rosenthal et al., 2015*) (but see [*Berdahl et al., 2013*]), or when they do consider animals moving in structured environments, by focusing only on social influences on movement (*Biro et al., 2006*; *Strandburg-Peshkin et al., 2015*; *King and Sueur, 2011*). Conversely, recent research in the growing area of movement ecology (*Nathan et al., 2008*) has yielded much insight into the internal factors and habitat-driven influences underlying animal movement patterns (*Coulon et al., 2008*;

*Hodson et al., 2010*; *Dickson et al., 2005*; *Leblond et al., 2010*; *Northrup et al., 2012*; *Mann et al., 2014*; *Johnson et al., 2015*; *McLean et al., 2016*), yet here the social drivers of movement are only beginning to be incorporated (*Vanak et al., 2013*; *Potts et al., 2014*; *Latombe et al., 2014*; *Basille et al., 2015*).

Habitat structure could alter collective movement patterns in a variety of ways. At the individual level, animals can exhibit goal-directed movement toward or away from certain habitat features (*Polansky et al., 2015*). They can also be constrained by physical features of the landscape such as steep hills or dense vegetation (*Wall et al., 2006*; *McLean et al., 2016*). At the collective level, these influences may result in changes to the organization, coordination, and movement patterns of entire groups. For example, theoretical models suggest that when some individuals within a collectively-moving group exhibit goal-directed movement, an elongated group structure can emerge (*Couzin et al., 2005*). The role of habitat structure could manifest itself both by altering the frequency with which groups exhibit characteristic structures (*Tunstrøm et al., 2013*), or by forcing the group into entirely new structures that would otherwise not occur. Habitat factors such as vegetation density could constrain the extent to which animals can coordinate their movement, both by limiting the flow of information (e.g. by reducing visibility to other group members) and by imposing physical constraints on where and how fast animals can move. Moreover, both individual movement rules and emergent group structure could be altered by environmental context, leading groups to be observed in different states or exhibit different behaviors depending on the area of their home range they occupy on a moment-to-moment basis. Thus, to understand how the varied patterns of group spatial organization and movement dynamics seen in nature emerge, taking the habitat into account is likely to be of critical importance.

Uncovering the role of habitat structure in shaping collective movement poses a considerable challenge, however, requiring simultaneous tracking of multiple individuals from the same social group combined with detailed information on their physical environment. Moreover, by contrast with studies assessing broad-scale habitat selection (*Dickson et al., 2005*; *Fortin et al., 2005*), in order to investigate decision-making at the local scale at which within-group social interactions occur, a finer-scale characterization of both the movements of animals and the environmental structure is required. In recent years, there has been an explosion of technological advances in both animal tracking (*Krause et al., 2013*; *Kays et al., 2015*) and remote sensing (*Anderson and Gaston, 2013*). Advances in GPS tracking capabilities now make it possible to collect data at a high enough spatial and temporal resolution that fine-scale movement decisions and social interactions can be resolved (*Nagy et al., 2010*; *King et al., 2012*; *Mann et al., 2014*; *Giuggioli et al., 2015*; *Strandburg-Peshkin et al., 2015*). At the same time, the increased consumer availability of unmanned aerial vehicles (UAVs), coupled with computational advances in image stitching and three-dimensional landscape reconstruction, have made these tools increasingly accessible to researchers in biology (*Anderson and Gaston, 2013*). Thus, we have only recently developed the capacity to address the influence of environmental heterogeneity on collective movement in the wild.

To investigate the interplay between social and habitat influences on collective movement at both an individual and a group scale, we studied the movement behavior and spatial organization of a troop of olive baboons (*Papio anubis*) within their natural habitat at Mpala Research Centre in Laikipia, Kenya. Baboon troops, which range in size from a few to over a hundred individuals, normally remain cohesive while foraging throughout the day, despite traveling long distances over variable routes through a heterogeneous savanna habitat (*Byrne, 2000*; *Cheney and Seyfarth, 2008*). Olive baboons are generalist omnivores, and typically feed on dispersed and widespread resources throughout their range. The two major predators of baboons are leopards and lions (*Cowlishaw, 1994*). Leopards, which will kill baboons both during the day and at night, are very common at the study site. To avoid predation at night, baboons choose safe sleeping sites and generally return to the same site each evening (although we observed our study troop temporarily changing sleeping site after the end of the study period after a leopard was observed at their sleeping site the previous night). Lions are much less common at the study site, and were unlikely to represent a major threat to our study group.

In a previous study (*Strandburg-Peshkin et al., 2015*), we used simultaneous high-resolution GPS tracking of individuals within a troop of wild baboons to investigate how baboons make movement decisions during daily travel in the context of the movements of their groupmates, revealing patterns consistent with a shared decision-making process governed by a majority rule. While analysis of

these movement data alone yielded insights into the social processes underlying collective movement, this work neglected the potentially important influence of habitat structure, both on the decisions of individuals and on the emergent collective behavior of the group.

Here we explicitly investigate the influence of habitat structure on collective movement by combining the above GPS tracking data of wild baboons (*Crofoot et al., 2015*) with fine-grained data on the habitat through which the group moved (*Figure 1A*, *Videos 1–2*). We collected high-resolution imagery of the baboons' habitat using an unmanned aerial vehicle (UAV), and from these images generated a 3-dimensional reconstruction of the habitat with c. 5 cm precision (*Appendix 1—figure 1*; *Appendix 1—figure 2*; *Appendix 1—figure 3*; *Appendix 1—figure 4*). Using model fitting procedures, we infer the relative importance of different social and habitat features in influencing the local (~5 m step length, see Materials and methods) movement decisions of individual baboons (*Appendix 1—figure 5*). We incorporate both small-scale and large-scale habitat features,

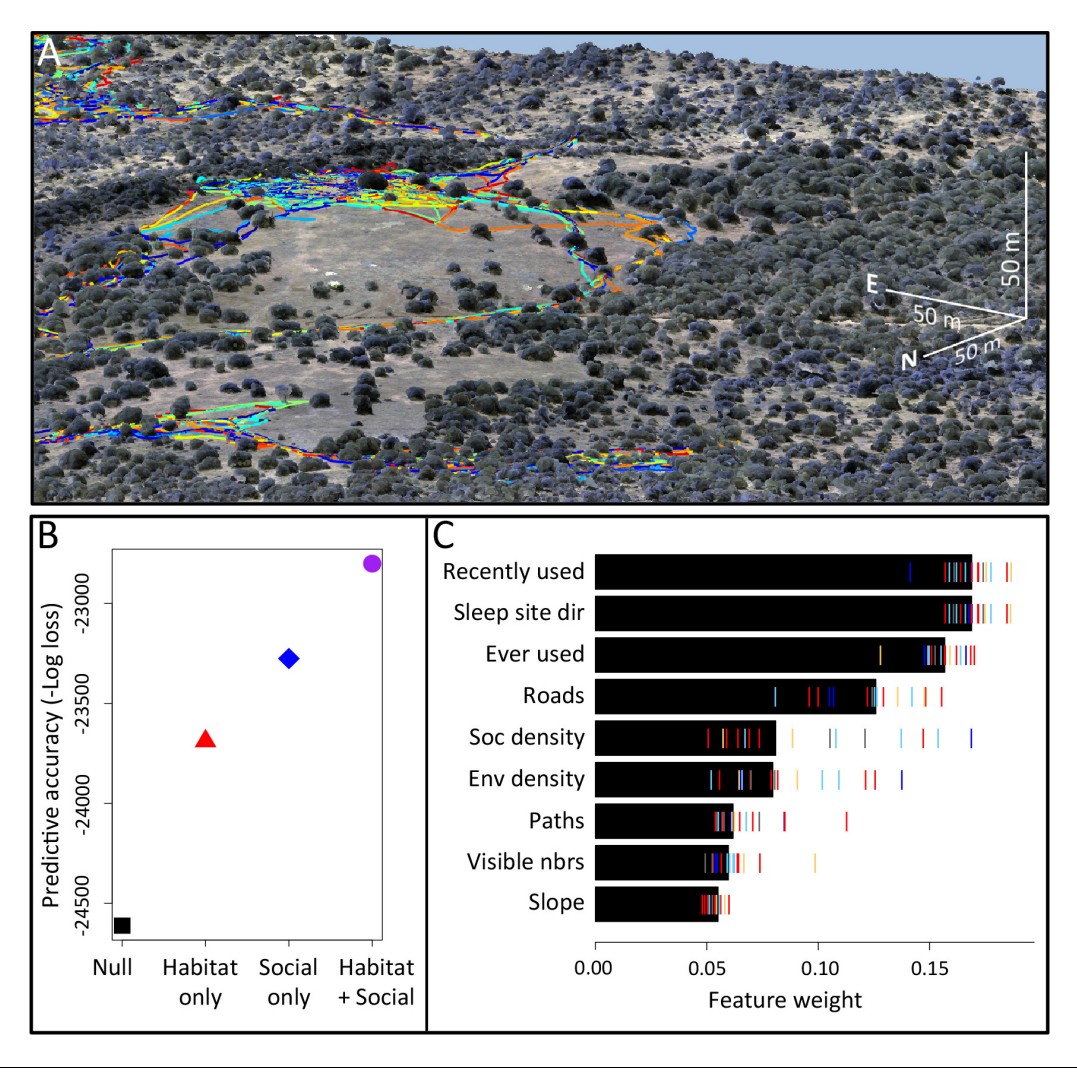

**Figure 1.** Combining three-dimensional habitat reconstruction with baboon tracking data to determine which habit and social features predict individual movement decisions. (A) Example of baboon trajectories overlaid on a high-resolution three-dimensional habitat reconstruction. Colored lines show trajectories of each baboon. (B) Predictive accuracy for step selection models using habitat features only (red point), social features only (blue point), or both social and habitat features (purple point), as compared to a null model (black point). Y-axis shows the negative log loss of out-of-sample data; points farther up the y-axis indicate better model predictions. (C) AIC weights associated with each feature based on multi-model inference. Each point shows the feature weights for a particular baboon individual, and black bars show the median feature weight across all individuals. Features are ranked from highest median feature weight (top) to lowest mean feature weight (bottom). See also *Videos 1–2*.

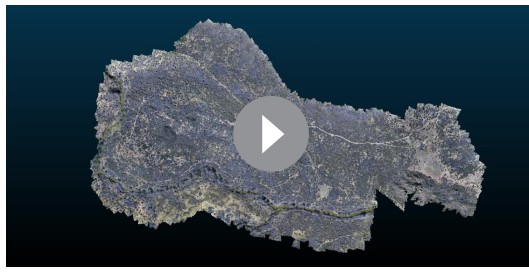

**Video 1.** Animation of 3-dimensional habitat reconstruction (point cloud data).

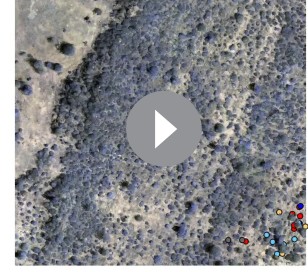

and both current and past positions of conspecifics into our models (see *Table 1* for a list of features included). We also use these data to quantify how the environment influenced emergent properties (e.g. shape and speed) of the baboon group, and to investigate how both individual-level decisions and group structure change in different environmental contexts.

**Video 2.** Example of baboon trajectories overlaid on 3-dimensional habitat reconstruction. Each point shows the movement of a single baboon within the troop, with color indicating the baboon's age and sex (red: adult female, dark blue: adult male, orange: subadult female, light blue: subadult male, gray: juvenile male).

## Results

### Which social and habitat drivers play a role in determining baboon movement decisions?

We used step selection models to evaluate the relative importance of social and habitat factors in influencing individual movement decisions. Step selection is a widely employed framework for assessing the environmental drivers of animal movement patterns (*Thurfjell et al., 2014*). Animal

**Table 1.** Features (predictor variables) used in conditional logistic regression models to predict baboon movement decisions.

| Feature | Description | Type |
|---|---|---|
| Environment density | Fraction of non-ground (vegetated) area within a 2.5 m* radius of a potential location | Habitat Feature |
| Social density | Fraction of all troop mates within a 4.25 m* radius of a potential location | Social Feature |
| Sleep site direction | Direction of a potential location relative to the sleep site, ranges from −1 (directly away) to 1 (directly toward), fit as interaction with time of day | Habitat Feature |
| Roads | Whether a potential location is on a road (1) or not (0). Fit as an interaction with whether the baboon's previous location was on a road | Habitat Feature |
| Recently-used space | Number of other baboons (not including focal individual) that have occupied a potential location within the past 4.5 min* | Social Feature |
| Ever-used space | Whether a potential location was ever occupied by another baboon (not including focal individual) across the entire dataset | Both |
| Animal paths | Whether a potential location is on an animal path (1) or not (0), fit as interaction with whether baboon's previous location was on a path | Habitat Feature |
| Visible neighbors | Fraction of other group members visible from a potential location (i.e. direct line-of-sight does not pass through any vegetated areas) | Social Feature |
| Slope | Change in elevation to a potential location from the baboon's previous location | Habitat Feature |

*Spatial and temporal scales were determined using maximum likelihood in a preliminary analysis. See Supplementary methods and *Appendix 1— table 1* for further details.

movements are considered as sets of discrete steps through a landscape, and probabilistic models are fitted to attempt to predict the next location an animal will move to out of a set of available options (herein *potential locations*), based on a set of features associated with each option. Alternative options are drawn from a distribution of step lengths (distance traveled during each step) and turning angles (angle between consecutive step directions), thus controlling for these basic properties of movement. The resulting model fits can thus be used to infer which environmental features are relevant to animal movement decisions, and their relative strengths of influence. Step selection models have previously been employed to infer the habitat preferences of animals (*Fortin et al., 2005*; *Coulon et al., 2008*; *Hodson et al., 2010*), as well as their responses to distributions of predators (*Fortin et al., 2005*; *Basille et al., 2015*; *Latombe et al., 2014*), and competitors (*Vanak et al., 2013*; *Potts et al., 2014*). However, to our knowledge step selection models have not before been applied to the study of the fine-scale social interactions within animal groups and their interplay with nonsocial influences on movement.

Here we identify a variety of both habitat and social features from our habitat and trajectory data (*Table 1*), and incorporate these as predictors into step selection models to determine which features are most important in predicting baboon local movement decisions. Because some features depend on the spatial or temporal scale over which they are measured, we also use maximum likelihood to infer from the data the most likely scales over which these features affect baboon movement decisions (see Materials and methods, *Appendix 1—figure 6–8*), following (*Mashintonio et al., 2014*). We fit separate step selection models for each baboon, enabling us to test whether the patterns we observe are general across all individuals.

Our analysis indicates that both social factors and habitat factors are important in determining baboon movement decisions. Comparing the predictive ability of models incorporating only social features (social only), only habitat features (habitat only), all features (habitat + social), or no features (null model) shows a clear increase in predictive power for models including both social and habitat features, as compared to those incorporating only one or the other (*Figure 1B*). To infer *which* habitat and social factors exert the greatest influence over baboon movement decisions, we compute the relative feature weight of each predictor using multi-model inference (*Figure 1C*; *Appendix 1— figure 5*; see Supplementary methods). We find that the top-ranked predictor (highest feature weight) of whether a baboon moved to a given location is the number of other baboons that had occupied that location within the recent past (*recently used* feature, *Figure 1C*). The best supported time scale associated with this effect is approximately 4.5 min (*Appendix 1—figure 8*); in other words, baboons show a preference to move to locations that have been occupied by many of their troop-mates within that timescale.

To confirm that this result is not an artifact of the fact that recently-used locations also represent locations that are suitable for baboons to move through, we include as a control in our models a binary predictor variable reflecting whether any baboon (other than the focal individual being modeled) ever used a particular location over the entire dataset (*ever used* feature, *Figure 1C*). We find that the *recently used* predictor achieves the top rank even when this control predictor is included, suggesting that locations recently used by conspecifics are attractive to baboons above and beyond the effect of physical suitability. It is also notable that this feature ranks much higher in the multi-model inference results than *social density*, which would be expected to rank highly if baboons were simply maintaining cohesion with other troop members.

In addition to *recently used* space, multi-model inference also reveals that habitat features influence movement, and that these operate at multiple spatial scales. The direction relative to the location of the sleep site emerges as an important predictor of individual movement decisions (*sleep site dir* feature ranked second, *Figure 1C*), with baboons tending to move away from the sleep site in the morning and towards it in the evening. Thus this spatial feature determines a general direction of travel for the group that is dynamic across the day. More locally, whether a location is on a road influences its probability of being selected (*roads* feature, *Figure 1C*), and in particular, baboons show a strong preference to follow roads once on them (*Appendix 1—figures 10–11*). By contrast, smaller game paths (*Appendix 1—figure 3*), although much more numerous than man-made roads, have a less pronounced influence on baboon movement decisions (*paths* feature is low-ranked, *Figure 1C*). Over a similarly local scale (2.5 m, *Appendix 1—figure 6*), baboons tend to avoid locations of high vegetation density (*env density* feature, *Figure 1C*). This feature, along with the change in elevation required to reach a location,

and the fraction of visible neighbors, plays a more minor role, but all are still supported by multi-model inference (*Figure 1C*). Thus, our model-fitting results support that baboons incorporate a variety of both habitat and social elements operating over a range of spatial scales when making movement decisions. From the model fits, we can also generate a data-driven 'preference

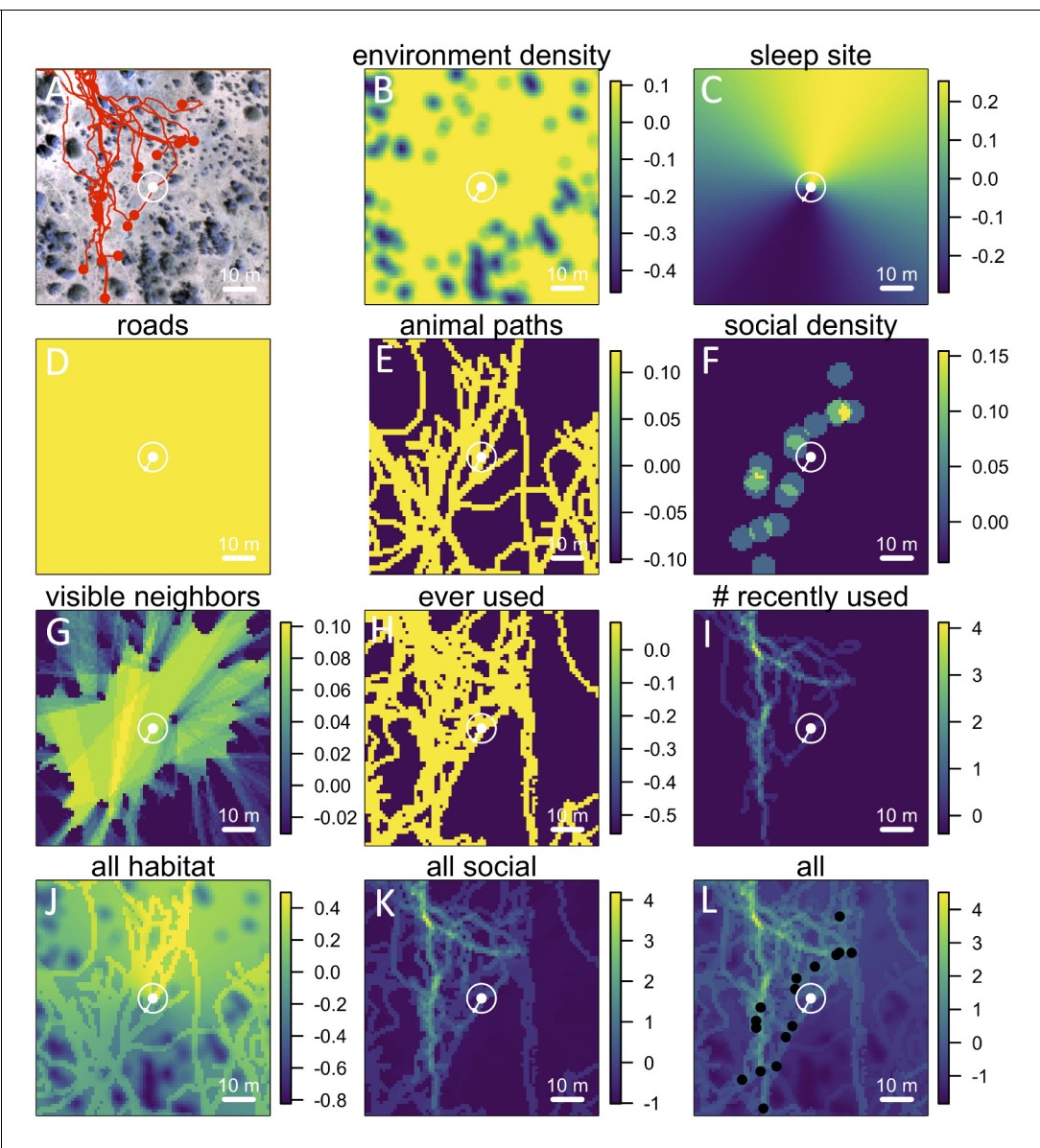

**Figure 2.** Visualizing the preference landscape underlying individual movement decisions. (A) Example of a single step taken by a focal individual. Background image shows overhead view of 3D habitat reconstruction. White marker shows the location of the focal baboon (whose next step is being modeled), and white circle shows a radius of 5 m (the specified step size) around the focal individual. White arrow shows the step actually taken by the individual. Red lines show recent locations of other baboons, and red points show locations of other baboons at the end of the step. (B –I) Visualizations of the influence of each habitat and social factor (the 'preference landscape') based on the fitted step selection model - lighter yellow areas represent locations that are more preferred, and darker blue areas are less preferred. Each panel represents the influence of a particular factor (ignoring all others) as predicted by the model: (B) vegetation density, (C) sleep site direction, (D) roads, (E) animal paths, (F) social density, (G) fraction of visible neighbors, (H) locations that have ever been used by another baboon, (I) number of baboons that have recently (in the past 4.5 minutes) used a location. (J–L) Last three panels represent overall preference landscapes, combining information from all habitat features only (J), all social features only (K), and all features, i.e. the full model prediction (L). For another example of preference landscapes (from a case where the focal individual started on a road), see *Appendix 1—figure 14*.

landscape' to visualize the influence of each of these features on individual movement decisions (*Figure 2*).

## How do social and habitat features combine to determine individual movement decisions?

In addition to determining which features are important, we also explicitly investigate the interplay between habitat and social features in influencing baboon movement decisions. To do so, we start by characterizing how baboon movement decisions are influenced by the previous locations of their troop mates (the top-ranked feature according to our multi-model inference results). We compare the properties of each real location where a baboon moved to those of an alternative location that it could have moved, with alternative locations generated by randomly drawing from the baboon's step length / turning angle distribution. For each pair of real and alternative locations, we randomly select one location and denote it *location 1*, with the other denoted *location 2*. We then compute the probability that *location 1* was the real location, as a function of the difference between the number of other baboons that had recently occupied *location 1* and the number that had recently occupied *location 2*. The result is a sigmoidal response (*Figure 3A*), where the true location selected by a baboon is likely to be the one that more of its troop-mates have recently passed through, with this probability increasing as the difference in the number of troop-mates to occupy the two alternative locations increases. This result is consistent with the previously-observed preference of baboons to follow the majority of troop-mates (*Strandburg-Peshkin et al., 2015*), but in this case the preference is shown to be for a particular location in space rather than a general direction of travel. Thus as baboons move through space they effectively 'pave the way' for their troop mates, who tend to follow in the footsteps of larger subgroups.

We then assess how this social influence on movement combines with habitat factors by testing whether the sigmoidal relationship is altered by the presence of roads (*Figure 3B–C*) and the sleep site direction (*Figure 3D–F*). When comparing two potential locations, if we subset our data to instances in which one potential location is on a road and the other is not, a clear bias towards choosing the on-road location emerges (*Figure 3B–C*, red lines), and is particularly prominent when the focal baboon starts on a road (*Figure 3B*). This result suggests that both roads and the previous locations of troop members strongly influence the locations where baboons choose to move, with the relative contributions of these two influences reflected in the shape of the sigmoidal curve (*Figure 3*). The sleep site direction is found to introduce a similar bias (*Figure 3D–F*), where in the morning baboons were more likely to choose locations farther away from the sleep site, and conversely for the evening. Taken together, these results illustrate how habitat influences on movement weigh against social influence. For example, from *Figure 3B* one can determine that when a baboon starts on a road, it takes 3–4 more troop mates to walk through an off-road location before a baboon will have an equal chance of picking it compared to an on-road location (indicated by the red line crossing y = 0.5).

The approach of computing probabilities directly from the data (*Figure 3*) is complementary to the model fitting approach described earlier: while model fitting allows us to control for many factors simultaneously, it also makes assumptions about the way in which these different factors combine to determine the probability of making a certain decision, whereas the direct approach allows us to relax these assumptions. Both approaches produce consistent results (see also *Appendix 1— figures 12–13*), giving us increased confidence as to their robustness, while also providing a clearer view of the interplay among different features affecting baboon movement decisions.

## How do individual movement decisions and emergent group structure vary across different environmental contexts?

By influencing the movement decisions of individuals, habitat structure can also shape movement patterns at the group level. Here we address this question by evaluating how group structure changes within different environmental contexts. We characterize group structure using six group-level metrics representing the global spatial organization and movement of the troop (*Figure 4*), and investigate four types of context: time of day (morning, midday, and evening), environment density (the fraction of the total area occupied by the group that is covered with vegetation), path density (fraction of group area that is on a path) and the presence of roads (whether a road is within the group area). In addition, to assess whether group-level shifts in structure could be driven by changes

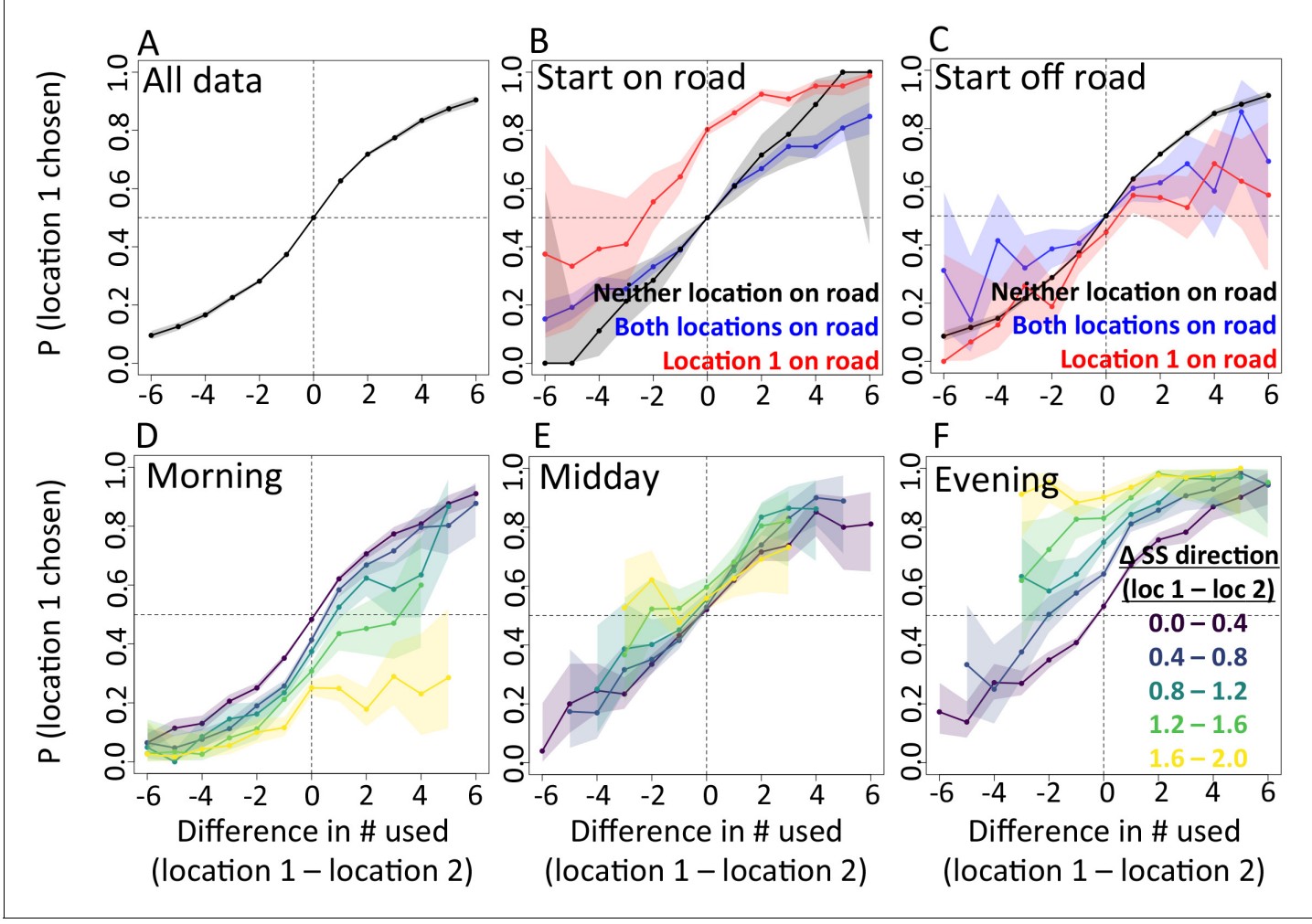

**Figure 3.** The interplay between habitat and social features in shaping individual movement decisions. We compare each real location where a baboon moved to an alternative location that it could have moved. We randomly select one of these two locations and denote it *location 1*, denoting the other *location 2*. Each plot shows the probability that *location 1* was the true location actually chosen by the baboon, as a function of the difference between the numbers of other baboons that had recently (within the past 4.5 min) occupied *location 1* and the number that had recently occupied *location 2*. (A) Across all data, the location chosen by the focal baboon is more likely to be the one recently occupied by more of its group mates. Moreover, the greater the difference between the number of baboons to have occupied each location, the stronger the effect (sigmoidal shape of curve). (B,C) Movement decisions are altered by the influence of roads. Here, data are shown from when the focal baboon started on a road (B) and when it started off a road (C), in three different cases: neither location was on a road (black line), both locations were on a road (blue line) or *location 1* was on a road whereas *location 2* was not (red line). (D–F) Movement decisions are influenced by the direction of the sleep site in the morning (D) and evening (F), but less in the midday (E). Each colored line shows data from instances when the difference in the steps' directedness toward the sleep site between *location 1* and *location 2* fell into a different bin (given in the legend), with lighter (more yellow) colors indicating a greater difference. When there is little difference (dark purple lines), the curve resembles that shown in panel A. As the difference increases, the location that is in the direction away from (in the morning) or towards (in the evening) the sleep site becomes more likely to be chosen by the baboon. Shaded regions denote 95% confidence intervals (based on Clopper-Pearson intervals). See also *Appendix 1—figure 12* (other environmental influences) and *Appendix 1—figure 13* (10 m steps rather than 5 m steps).

in individual-level movement priorities, we fit separate step selection models for the data within each context and compare the feature ranks across these different contexts.

Considering different contexts reveals broad consistency, but also telling changes, in the priorities driving baboon movement at the individual level (*Appendix 1—figure 15*; *Appendix 1—figure 16*). For example, models fitted using data that correspond to different times of day demonstrate that the importance of roads is stronger in the morning and evening relative to the middle of the day. The direction of the sleep site also plays a larger role in the morning and evening than in the midday.

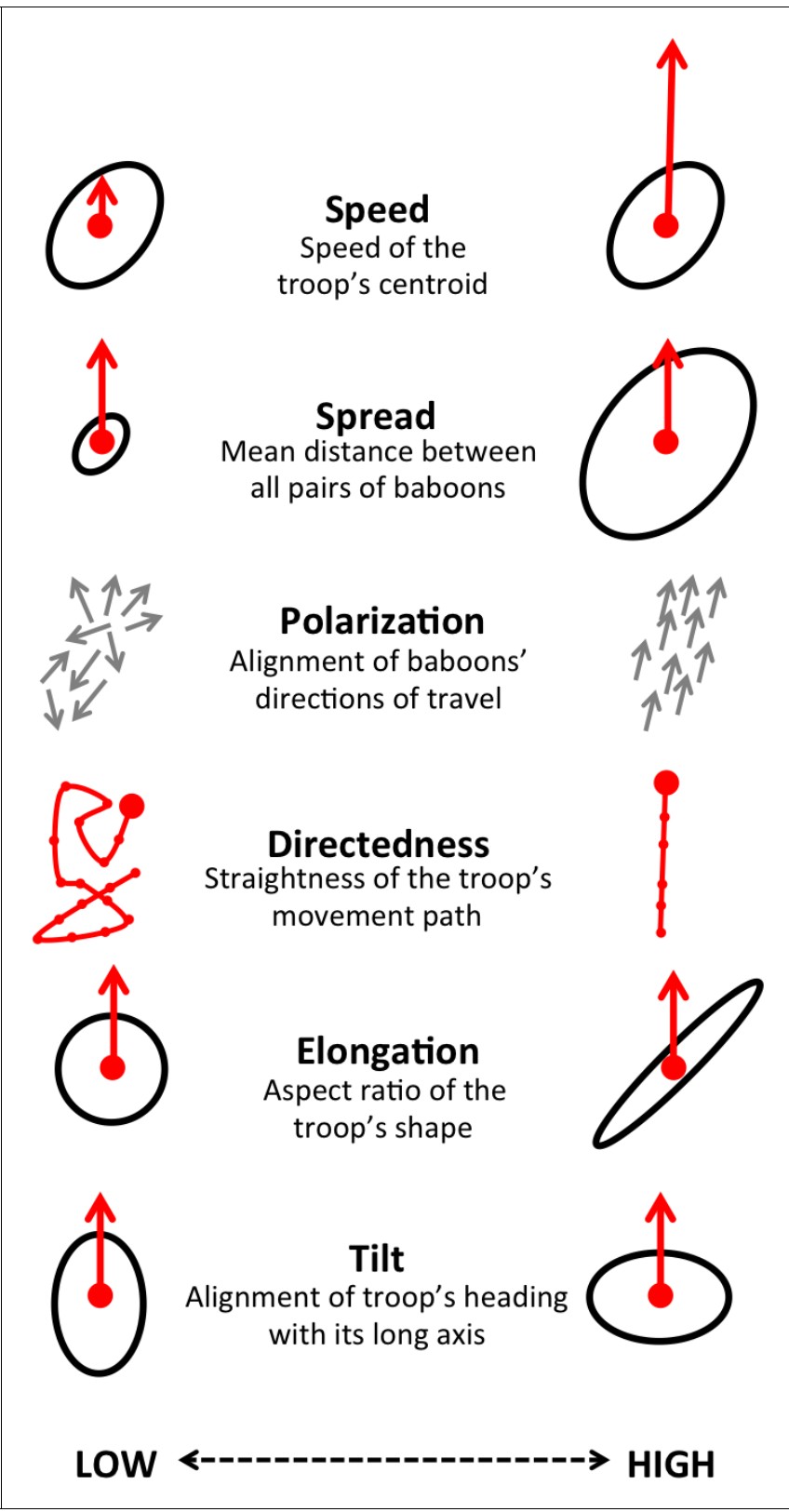

**Figure 4.** Illustration of the six measures used to characterize group-level properties. For further information, see Supplementary methods.

This suggests that baboons are effectively 'commuting' to and from foraging locations during the start and end of each day (*Appendix 1—figure 15A*). These individual-level changes are coupled with changes to group structure at the collective level . During morning and evening, the troop tends to be more elongated, and to move more often in a directed manner with lower tilt (*Appendix 1—figure 18*, *Appendix 1—figure 22*), all of which would be expected based on either more goal-directed movement (toward or away from the sleep site) and/or movement that utilizes roads. Indeed, the group spent considerably more time on roads during morning and evening than during the midday (12.1% in morning and evening vs. 7.5% during midday, *Appendix 1—figure 23*). Moreover, comparing group structure on and off roads supports the idea that baboons use roads as an efficient method of travel, with groups moving faster and in a more directed manner on roads (*Figure 5B*, *Appendix 1—figure 20*).

In addition to large-scale habitat features such as roads and the sleep site, individual movements are also constrained by fine-scale vegetation structure, with consequences for the emergent structure and coordination of groups. Although when data are considered in aggregate, environment density plays a relatively small role in influencing the local movement decisions of individual baboons (*Figure 1C*), its importance increases when considering individual movement decisions in particularly dense environments, surpassing both *roads* and *social density* (*Appendix 1—figure 15B*). Thus, in dense environments, baboons' movement decisions become more strongly influenced by local-scale heterogeneity in vegetation relative to other factors. At the group level, high-density areas are associated with lower group polarization, as well as slower troop speed and less directed travel (*Figure 5C*). These changes suggest that vegetation structure plays an important role in constraining the degree of movement coordination in animal groups. Similarly, though animal *paths* are low ranked in feature weight when models are fitted across all data, their importance at the individual level increases in areas with a particularly high density of paths (*Appendix 1—figure 15C*). Areas

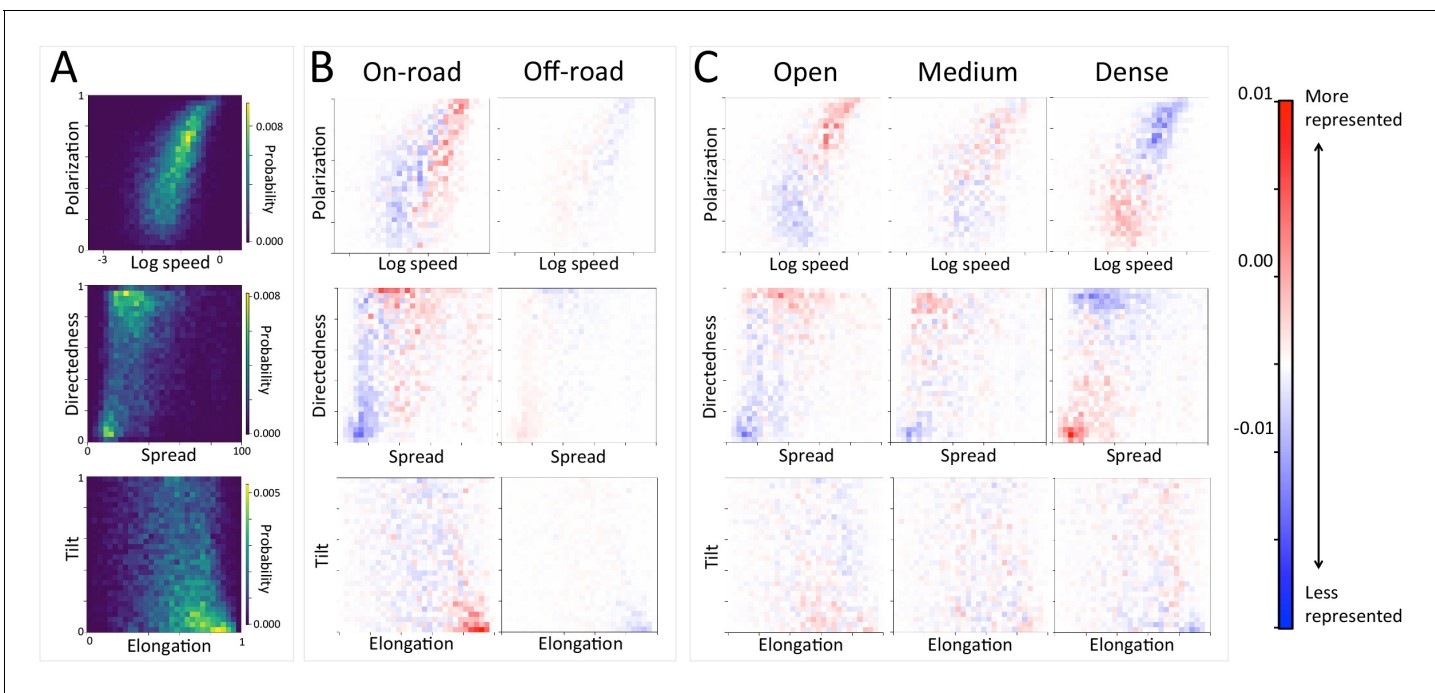

**Figure 5.** At the group level, troop structure and movement changes based on roads and vegetation density. (A) Two-dimensional histograms of the speed and polarization (top panel), spread and directedness (middle panel), and elongation and tilt (bottom panel) of the baboon troop across all data. Lighter areas represent more likely group configurations. (B–C) Difference between the distributions within a given context and the overall distribution across all data. Redder areas represent group configurations that are over-represented within a given context (relative to the rest of the data), and bluer areas represent under-represented configurations. Each column represents a different context: (B) on-road and off-road, and (C) open, medium, and dense vegetation. See also *Appendix 1—figures 17–18* for the influence of path density and time of day respectively, as well as *Appendix 1—figures 19–22* for one-dimensional histograms of each group-level property within each context.

with a high density of paths are also associated with faster, more directed, and more polarized movement at the group level (*Appendix 1—figure 17*, *Appendix 1—figure 21*).

To disentangle the effects of different habitat factors and time of day on group-level properties, we fit linear models predicting each group-level property as a function of habitat density, path density, and presence of roads within the area occupied the group, as well as time of day (morning, midday, and evening). Computing the feature weights of each of these predictors via multi-model inference reveals that the habitat features receive larger weights across all group-level properties than does time of day (*Figure 6*). This result suggests that group structure is shaped more by the current habitat the baboons occupy than by the temporal context. The fits also reveal that roads and areas of high path density are associated with similar changes to group structure – i.e. groups become faster, more elongated, and more polarized – but that roads are generally a stronger predictor of these changes (higher feature weight, *Figure 6*). By contrast, in dense habitats essentially the reverse of this pattern is seen.

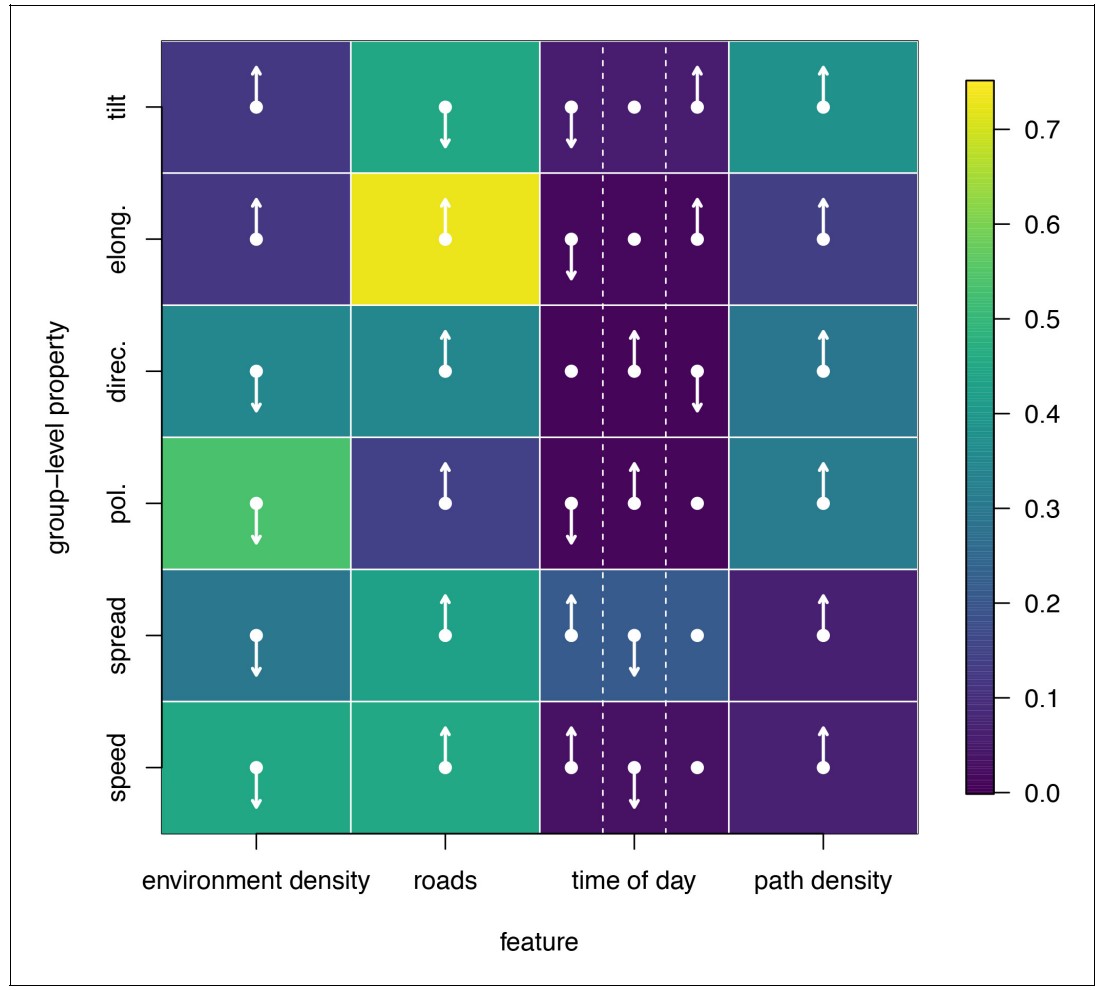

**Figure 6.** Results of fitting linear models to predict group-level properties as a function of features of the habitat occupied by the group and the time of day. Colors show feature weights, with lighter colors indicating habitat/temporal features (columns) that were more supported by multi-model inference to predict each group-level property (rows). Arrows indicate the direction of the effects for each fit, with upward (downward) pointing arrows indicating a positive (negative) effect. For time of day column, arrows correspond to morning (left), midday (middle), and evening (right), with the upward (downward) pointing arrow indicating the time of day associated with the largest (smallest) value of the group-level property in each row.

# Discussion

Efforts to understand collective animal movement (*Sumpter, 2010*) and habitat influences on individual movement (*Boyce and McDonald, 1999*; *Nathan et al., 2008*; *Thurfjell et al., 2014*) have constituted major avenues of research in recent years. It is perhaps surprising, then, that these parallel streams of effort, which both aim to understand and quantify the patterns of animal distribution and movement observed in nature, have so far had relatively little interaction with one another (but see [*Potts et al., 2014*; *Berdahl et al., 2013*]). By exploiting recent advances in GPS tracking, remote sensing, and analytical methods, we uncover a range of both habitat and social influences, operating over varied spatial scales, which exert influence on the local movement decisions made by individual baboons and ultimately the aggregate structure and dynamics of the group. Changes in coordination and group spatial organization as a function of environmental factors will have important consequences for understanding both the mechanistic basis of group coordination and its costs and benefits.

We demonstrate that a variety of habitat features wield influence on individual movement decisions. Moreover, habitat influences extend beyond the level of individual decisions to the collective dynamics of the group, leading to changes in emergent group structure and coordination in different environmental contexts. This effect is manifested in our dataset in the fast movement and elongated group structures formed on roads and areas of high path density, and the decrease in coordination when moving through dense habitats. These changes could reflect direct habitat constraints on movement (such as physical barriers to movement), but also behavioral changes associated with different habitats (such as a shift from foraging to directed travel in different environments).

Our results suggest that, although few in number and in total area, roads and sleeping sites are relatively dramatic features of the habitat that have a large impact on baboon movement patterns. The importance of roads is consistent with studies in other species which suggest that roads may facilitate animal movement (*Dickson et al., 2005*), perhaps by serving as navigational landmarks (*Mann et al., 2014*). This hypothesis is consistent with the weaker influence of paths in our analysis (*Figure 1C*), whose sheer abundance and tortuosity could make them less useful as navigational aids. We also found that roads were more important predictors of baboon movement in the mornings and afternoons than during the middle of the day. This temporal difference is likely to reflect the needs of the troop to travel away from their core zone (near their sleeping site) to forage (*Appendix 1—figure 23*). In fact, baboons in our troop often travelled 5–6 kms per day during the study period, probably because conditions were dry and food, although not scarce, was not abundant enough close to their sleep site to sustain the entire troop.

Our analysis also uncovers a primary role of social interactions - in particular a baboon's tendency to follow at a very fine scale where others have gone before - in shaping individual movement decisions. Baboons' tendency to follow in the footsteps of others emerges as the highest-ranked predictor of individual movement decisions in our analysis. There are several possible explanations for this finding. The first possibility is that individuals are following others to resource patches within the landscape. However, in order for this effect to be driving our result, our study troop would have to be foraging on patches that are both ephemeral (so that they would not be accounted for by the *ever used* feature we included as a control) and slowly depleting (so that followers would benefit from visiting locations where others previously foraged). Our observation in the field is that the troop forages on mixed resources that are widely distributed (e.g. grass roots and prickly pear fruit), meaning that this explanation seems unlikely. Alternatively, the importance of following others could reflect a risk reduction strategy, since locations that conspecifics have recently passed through have already been effectively 'pre-screened' for threats such as leopards or venomous snakes. This hypothesis is supported by the decrease in the importance rank of the *recently used* feature when the troop is in open habitats (*Appendix 1—figure 15B*), as there are likely to be fewer anti-predator benefits to be gained by following others in open areas. A third possibility is that baboons moving through the landscape trample local vegetation, in turn making recently-used paths easier to follow. However, if this were the case, then we would expect existing animal paths to also emerge as important predictors of movement, which in our case they did not (*Figure 1C*). A final possibility is that since following in the footsteps of others is a less cognitively demanding task than carving out one's own path (*Piyapong et al., 2007*), by generally following these paths baboons are reducing the cognitive effort of navigation (*Kool et al., 2010*). Future work, including explicit measurement of both the resource distribution and the behavior that individuals engaged in while moving could help

discriminate among these possibilities. Regardless of the explanation, baboons' tendency to follow in the footsteps of others is likely to have consequences for the aggregate space use patterns of baboon troops, with troops making use of a smaller portion of the area within their surroundings than would be naively expected if individuals were moving more randomly within the troop's boundaries, which in turn could alter their impact on the environment around them.

Our results also have implications for understanding the costs and benefits of within-group spatial positioning. Where individuals position themselves within groups is thought to reflect a trade-off between forging new ground to find resources and being exposed to predators (*Hirsch, 2007*; *Krause, 1994*; *Parrish and Edelstein-Keshet, 1999*; *Hirsch, 2011*; *Teichroeb et al., 2015*). While work in this area has typically focused on how this tradeoff can drive differences in the amount of time individuals spend at the front vs back of the group, or in the center vs periphery, our results suggest that the extent to which animals follow in each others footsteps is another dynamic which should be considered, as this factor will have an important impact on the rate at which individuals are exposed to new areas - and hence the rate at which they will encounter both new resources and stationary predators.

While our work illustrates the insight that can be gained by combining step-selection models with high-resolution collective movement and environment data, we were limited to a relatively short period of data collection (weeks) for logistical reasons, largely relating to limited battery life. An exciting future direction would be to conduct longer-term or repeated sampling of a troop across the year to capture changing environmental conditions, and to investigate how these changes impact both individual decisions and group-level properties. For example, when conditions are very dry or very wet, do individual movement rules change, and do these conditions impact group structure? Wet conditions are likely to lead to more food, but also denser habitat and therefore more risk from ambush predators such as leopards. If predation is the primary reason individuals tend to move to locations where others have recently been, these conditions should increase the propensity for individuals to carefully follow others. Further, our study only accounted for short-term social drivers of movement. Recent theoretical (*Bode et al., 2011*) and empirical (*King et al., 2008*; *King and Sueur, 2011*) studies have highlighted potential for differences in social relationships to affect movement. Independent observations of affiliations, notably grooming relationships, could be also incorporated into our model to explicitly test when and how these inter-individual relationships are important for movement.

While the results shown here are specific to the system we explore, the methods employed and many of the insights gained are likely to extend to a variety of social animals. As the resolution at which we can map both the movements of animals and their environments improve, the integration of habitat features into studies of collective behavior - and conversely the incorporation of within-group social interactions into studies in movement ecology - will be critical for understanding the movement patterns of social animals.

## Materials and methods

GPS tracking data (1 Hz) were collected from 25 wild olive baboons (*Papio anubis*) belonging to a single troop at Mpala Research Centre in Laikipia, Kenya, from 1 August - 2 September, 2012. Tracked individuals included 81% (23/ 29) of the adult and subadult members of the troop, as well as two juveniles (for a total of 25). We did not apply any smoothing to the GPS data, but filled in the few short (1 s) gaps in the data using linear interpolation, as well as filtering out a small number of outlier data points (see Supplementary methods). Aerial imagery of the habitat was collected from 9–17 January, 2015 using an eBee fixed-wing UAV (Sensefly Ltd). The mapped area covered 3.3 km$^2$, representing approximately 2/3 of the area covered by the baboon troop during the first 14 days (the period from which we analyzed tracking data). Although we had a gap between the two data collection periods, both had similar ecological conditions (dry season) and habitat characteristics (trees, roads, bushes) were unlikely to have changed much over this period. After collection, images were automatically combined using Postflight Terra 3D software (Pix4D) to reconstruct a three-dimensional point cloud image of the landscape, and these were used to identify all habitat features. All procedures were subject to ethical review and were carried out in accordance with the approved guidelines set out by the National Commission for Science, Technology and Innovation of

the Republic of Kenya (NACOSTI/P/15/5727/4608). Baboon tracking was approved by the Smithsonian Tropical Research Institute (IACUC 2012.0601.2015).

We used a step selection framework to determine the habitat and social features most predictive of baboon movement decisions. To determine which habitat and social features (*Table 1*) were important in the fitted conditional logistic regression models, we used multi-model inference and computed the AIC weights of each feature. We quantified the group structure at each minute using six troop-level metrics (*Figure 4*). To characterize how group spatial structure and movement depended on context, we compared distributions of the metrics across different contexts by subtracting the histogram computed using all data from the histogram computed using data within each context. This allowed us to determine which areas of parameter space were over- and underrepresented in a given context. We also fit linear models predicting each group-level property as a function of all possible combinations of habitat and temporal contexts, then computed AIC weights to determine the relative importance of each of these contexts in predicting each group-level property. See Supplementary methods for further details.

## Acknowledgements

We thank Kenya National Science and Technology Council, Kenyan Wildlife Service and Mpala Research Centre for permission to conduct research. We thank M Wikelski, E Bermingham, D Rubenstein, M Kinnaird, D Carlino, and B Tripard for logistical support; R Kays, S Murray, M Mutinda, R Lessnau, S Alavi, J Nairobi, F Kuemmeth, W Heidrich, T Berger-Wolf, D Pappano, I Brugere, and J Li for field assistance; T Berger-Wolf, B Ziebart, I Brugere, J Li, A Scharf, B Kranstauber, K Safi, C Twomey, and S Garnier for helpful discussions; and members of the Farine lab for comments on the manuscript. We acknowledge funding from the Max Planck Institute for Ornithology, the Smithsonian Tropical Research Institute, and Princeton University. ASP was supported by an NSF GRFP fellowship, NIH 5T32HG003284-12, and a Charlotte Elizabeth Proctor Fellowship. DRF received additional funding from the BBSRC (BB/L006081/1 to BC Sheldon). MCC received support from NSF BCS 1440755 and III 1514174. IDC acknowledges support from NSF (PHY-0848755, IOS-1355061, EAGER-IOS-1251585), ONR (N00014-09-1-1074, N00014-14-1-0635), ARO (W911NG-11-1-0385, W911NF-14-1-0431), and Human Frontier Science Program (RGP0065/2012).

## Additional information

### Competing interests

IDC: Reviewing editor, *eLife*. The other authors declare that no competing interests exist.

### Funding

| Funder | Grant reference number | Author |
| --- | --- | --- |
| National Science Foundation | GRFP fellowship | Ariana Strandburg-Peshkin |
| Princeton University | Charlotte Elizabeth Proctor 335 Fellowship | Ariana Strandburg-Peshkin |
| Max Planck Institute for Ornithology | | Damien R Farine<br>Iain D Couzin |
| Biotechnology and Biological Sciences Research Council | BB/L006081/1 | Damien R Farine |
| National Science Foundation | III 1514174 | Margaret C Crofoot |
| National Science Foundation | BCS 1440755 | Margaret C Crofoot |
| Smithsonian Tropical Research Institute | | Margaret C Crofoot |
| National Science Foundation | IOS-1250895 | Margaret C Crofoot |
| National Science Foundation | EAGER-IOS-1251585 | Iain D Couzin |
| National Science Foundation | IOS-1355061 | Iain D Couzin |
| National Science Foundation | PHY-0848755 | Iain D Couzin |

| Army Research Office | W911NG-11-1-0385 | Iain D Couzin |
| Office of Naval Research | N00014-091-1074 | Iain D Couzin |
| Human Frontier Science Program | RGP0065/2012 | Iain D Couzin |
| Army Research Office | W911NF-14-1-0431 | Iain D Couzin |
| Office of Naval Research | N00014-14-1-0635 | Iain D Couzin |

The funders had no role in study design, data collection and interpretation, or the decision to submit the work for publication.

### Author ORCIDs

Ariana Strandburg-Peshkin, http://orcid.org/0000-0003-2985-6788
Damien R Farine, http://orcid.org/0000-0003-2208-7613

### Ethics

Animal experimentation: All procedures were subject to ethical review and were carried out in accordance with the approved guidelines set out by the National Commission for Science, Technology and Innovation of the Republic of Kenya (NACOSTI/P/15/5727/4608). Baboon tracking was approved by the Smithsonian Tropical Research Institute (IACUC 2012.0601.2015).

### Author contributions

AS-P, Acquisition of data, Analysis and interpretation of data, Drafting or revising the article; DRF, MCC, IDC, Acquisition of data, Drafting or revising the article

## Additional files

### Major datasets

The following dataset was generated:

| Author(s) | Year | Dataset title | Dataset URL | Database, license, and accessibility information |
| --- | --- | --- | --- | --- |
| Strandburg-Peshkin A, Farine DR, Crofoot MC, Couzin ID | 2015 | Data from: Habitat structure shapes individual decisions and emergent group structure in collectively moving wild baboons | http://dx.doi.org/10.5061/dryad.6h5b7 | Available at Dryad Digital Repository under a CC0 Public Domain Dedication |

The following previously published dataset was used:

| Author(s) | Year | Dataset title | Dataset URL | Database, license, and accessibility information |
| --- | --- | --- | --- | --- |
| Crofoot MC, Kays R, Wikelski M | 2015 | Data from: Shared decision-making drives collective movement in wild baboons | https://www.datarepository.movebank.org/handle/10255/move.405 | Data publicly available at Movebank Data Repository (www.datarepository.movebank.org) |

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

## Appendix 1

### Appendix Summary

This appendix contains additional information on methods (*Appendix 1—figures 1–5*) and results of additional analyses supplementing the findings presented in the main text (*Appendix 1—figures 6–23*).

### Supplementary methods

#### Collection and pre-processing of baboon movement data

From 21–29 July, 2012, we captured 33 out of the 46 members of a wild troop of olive baboons (*Papio anubis*) using individual (1 m$^2$) traps baited with maize. Seven baboons were too small to fit a collar, and were released immediately. Each of the remaining 26 individuals was chemically immobilized using Ketamine (15 mg/kg) and fitted with a GPS collar (e-Obs Digital Telemetry, Gruenwald, Germany) weighing less than 5% of its body weight. D-cell battery collars (300 g) were used for larger individuals, and C-cell battery collars (230 g) were used for smaller individuals. The collars were programmed to record baboon locations continuously at a rate of 1 Hz during daylight hours (06–18 hr). Because the collars sampled data continuously, position error was low. During a test walk in which collars were held 1 m apart, the average relative error was found to be <0.26 m (95% CI: 0.03–0.69). One collar failed immediately, therefore we analyzed the data from the remaining 25 collared individuals. Due to the limited battery life of the C-cell collars, we analyzed only the first 14 days of collar deployment, when most collars were active. See *Strandburg-Peshkin et al. (2015)* for more information on collar deployment, including the recording duration for each individual.

We did not apply any smoothing to the GPS data. Occasionally, a device failed to log one or a few points, and in these cases we estimated the missing values using linear interpolation. This procedure was found to result in an error of less than 0.2 m (less than the GPS error), based on an analysis in which we randomly removed 5000 existing data points and compared their values to those estimated using linear interpolation. In addition, we found a few instances of unrealistic speeds in the data, suggesting GPS errors. Using the same linear interpolation algorithm, we identified points above the 99.9th percentile error between interpolated and observed data points (equivalent to speeds above 3.56 m/s, and replaced these with their linearly interpolated values. Overall, 7.4% of the data used was interpolated, with 7.2% of these being replacements of missing values.

#### Collection and pre-processing of habitat data

##### Collection of aerial imagery

To collect aerial imagery data, we deployed an eBee fixed-wing UAV (Sensefly Ltd). The eBee was deployed at heights of approximately 120 m above the ground, achieving image resolution of approximately 5 cm per pixel. Note that aerial imagery was collected approximately 2.5 years after GPS tracking data were collected. Although the small-scale vegetation structure may have changed during this time period, the general positions of landscape features such as the locations of acacia trees, roads, etc. are unlikely to have dramatically changed. In addition, both GPS data and aerial imagery were collected during dry seasons, suggesting that overall vegetation density was likely to be similar.

## Ground classification

Each point in the 3D point cloud data images obtained from the Postflight software was automatically classified as ground or non-ground (*Appendix 1—figure 2B*) using the open-source Multiscale Curvature Classification for LIDAR Data (MCC-LIDAR) software (*Evans and Hudak, 2007*). MCC-LIDAR uses an algorithm based on the local curvature to differentiate areas in a point cloud image that are likely to be ground vs. non-ground objects. The algorithm has two parameters: the scale parameter (set at 1.5 here) and the threshold parameter (set at 0.3 m here). While visual inspection of the results confirmed that the ground classification algorithm had performed well on the majority of the mapped area, the algorithm often misclassified regions very near the river. All data within 50 m of the river were therefore excluded from our analyses. Because the sleep site was on the river, this also ensured that data from when the group was very near to the sleep site were not included in the analysis. Similarly, because the edges of the mapped region tended to be less reliably imaged, all data within 20 m of the map edge were excluded.

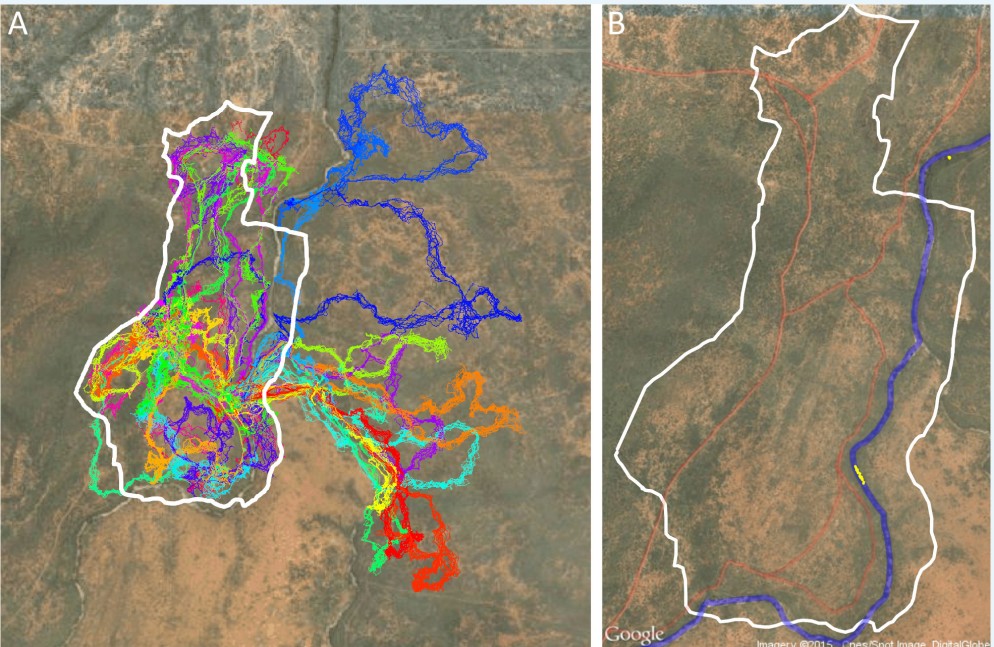

**Appendix 1—figure 1.** Drone-mapped area and identification of roads and sleep site locations. (**A**) UAV-mapped area in relation to troop range during the entire period of tracking. Each thin colored line represents the daily trajectory of a single baboon, and different colors represent different days (though 35 days of tracking data are shown here, only the first 14 days of data were used in the analysis). Thick white lines enclose the UAV-mapped region, for which we generated a three-dimensional habitat reconstruction (see *Figure 1A*). (**B**) Map showing the locations of roads (red lines), the river (blue line), and the troop's sleep site (yellow circles indicate the troop's sleeping location on each morning, defined as the centroid of all baboon locations).

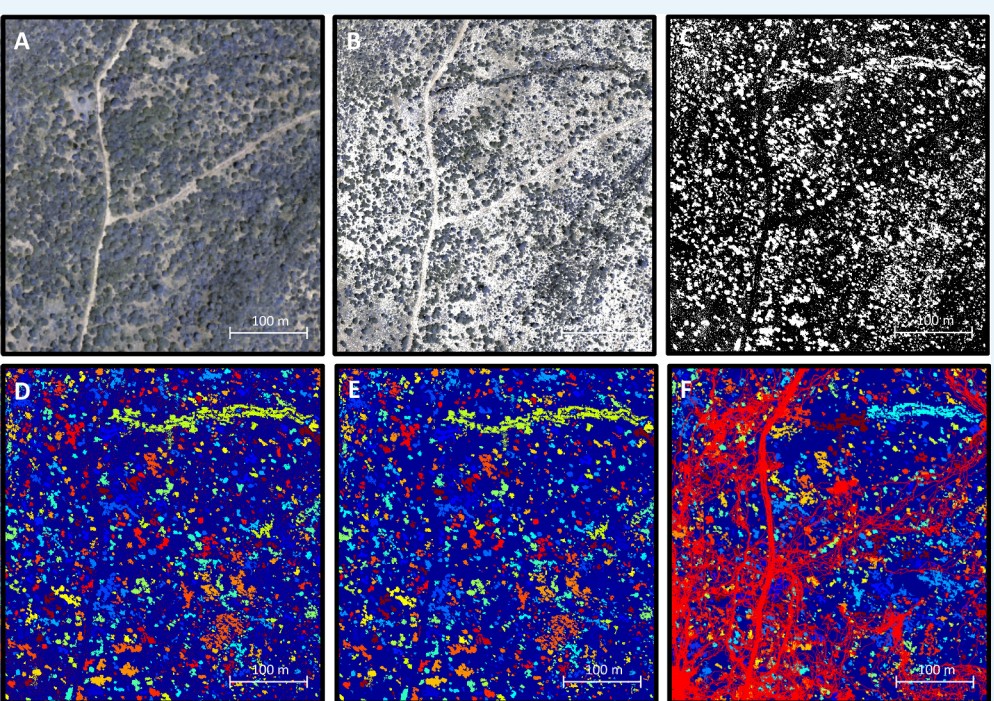

**Appendix 1—figure 2.** Processing 3-dimensional habitat reconstruction. Starting with the 3-dimensional habitat reconstruction data (**A**), we used a ground the classification algorithm MCC-LIDAR to identify non-ground points (**B**). We then rasterized these data to a grid with 20 × 20 cm pixels (**C**). Here, black pixels indicate ground and white pixels indicate non-ground. Using image segmentation, we identified connected clusters of pixels (**D**), here represented as sections of different colors (with the ground here represented in blue). We then removed all identified objects smaller than 2 $m^2$ to reduce small-scale noise (**E**). (**F**) shows the baboon trajectories (in red) overlaid on the processed image.

## Rasterizing data

The 3D point cloud data were rasterized onto a two-dimensional binary grid (*Appendix 1—figure 2C*). Each 20 × 20 cm grid cell was defined as 1 if there was at least one non-ground point within its range (regardless of the height of this point), and 0 otherwise. Thus, we simplified the three-dimensional information of the point cloud data to a two-dimensional representation, indicating whether a particular 20 × 20 cm area was occupied or not occupied by vegetation (non-ground structure primarily consisted of vegetation, though large rocks, etc. could also be incorporated into this measure).

The image was then segmented to separate out distinct objects (connected components) using the bwconncomp function in MATLAB (*Appendix 1—figure 2D*). These objects likely represent trees, large rocks, etc. To reduce small-scale noise, objects smaller than 2 $m^2$ (50 pixels) in area were removed (the grid cells they occupied were set to 0; *Appendix 1—figure 2E*). A section of the final environment raster, with the small objects removed, and with baboon tracks overlaid, is shown in *Appendix 1—figure 2F*.

## Identifying roads

All roads were identified by tracing them out manually using Google Earth Pro (*Appendix 1—figure 1B*, red lines). The area within 3 m of the center of the road was considered to be 'road' in subsequent analyses.

## Identifying the sleep site

The sleep site was identified on each day as the first tracked location of the group. This was done by finding the first time on each day where at least 10 individuals were tracked, then taking the centroid (average position) of those 10 individuals at that time. Since GPS collars turned on at 6 am each day, the group was always located at its sleep site (typically in a particular set of trees along the river) at this time. Sleep site locations for each day are shown as yellow points in *Appendix 1—figure 1B*. Note that the sleep site remained consistent across all 14 days used in the analysis. In fact, the baboons did switch to an alternate sleep site near the very end of data collection (yellow points on river in north), but because this occurred after the first 14 days it was not included in our analysis.

## Identifying animal paths

Animal paths (*Appendix 1—figure 3*) were traced out manually by six paid research assistants who did not have access to the baboon trajectory data and were otherwise not involved in the project. Georeferenced orthomosaic images were imported into Google Earth Pro and converted into superoverlay files, allowing the assistants to zoom in and out in the image files to enable better visualization. Paths were traced out by clicking on each consecutive location in the path using Google Earth Pro's path creation feature. Each assistant traced out paths for the whole dataset independently. Because paths were so numerous that no student was able to label all of them in the allotted time, paths from all students were combined into a single dataset. For the purposes of analysis, regions of space were classified as on a path if they were within 1 m of a path traced by any of the students, and not on a path otherwise.

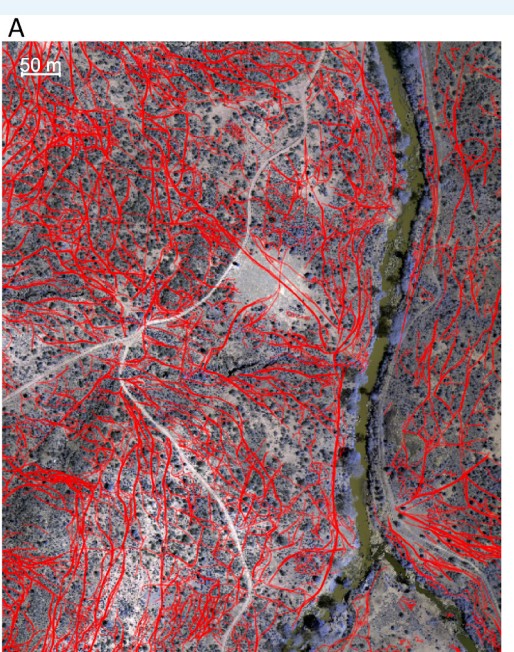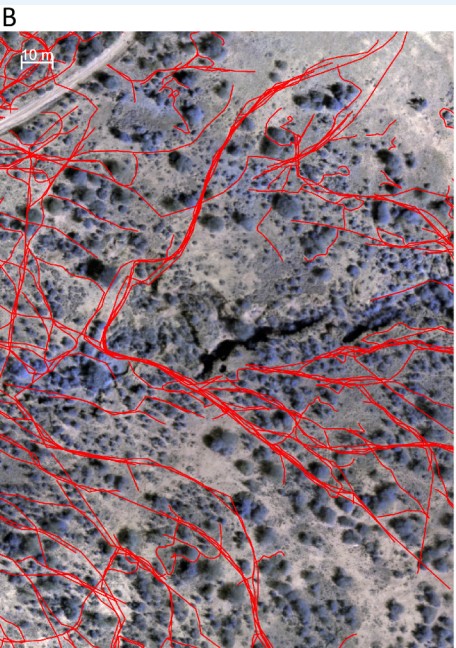

**Appendix 1—figure 3.** Identifying animal paths. Animal paths, shown in red here, were traced out manually from the orthomosaic images using Google Earth Pro. (**A**) and (**B**) show a large area and zoomed in region, respectively.

## Identifying the fraction of visible neighbors

For each location in space at a given time, we were able to estimate the fraction of other baboons (i.e. those other than the focal baboon being modeled) that were visible from this vantage point (*Appendix 1—figure 4*). For each other baboon, we determined whether the

direct line-of-sight from that location was clear (i.e. a line drawn from the location to the baboon was not blocked by vegetation or other non-ground objects), and we then computed the fraction of clear lines-of-sight across all other baboons. A line-of-sight was defined as blocked if it intersected with any non-ground pixels in the raster. This measure is therefore likely to somewhat underestimate the visibility, since some smaller objects that baboons might be able to see over could still be counted as blocking the view. However, our analysis did reduce this effect by removing very small-scale objects from the raster (see section above, and *Appendix 1—figure 2E*).

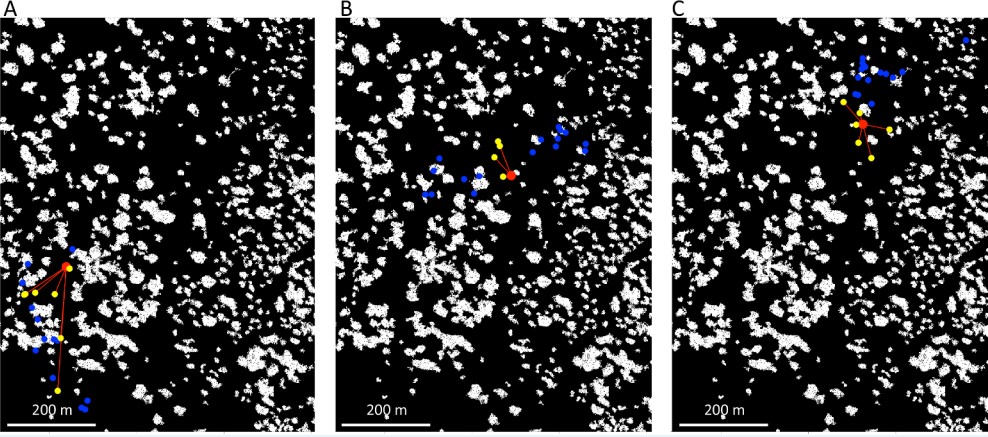

**Appendix 1—figure 4.** Estimating which baboons are visible from a given location. We estimated the direct line-of-sight from a given focal location (red marker), assuming that if this line-of-sight crossed through any non-ground pixels, it was blocked. Here, yellow markers show baboons that are visible from the focal location, with red lines connecting them. Blue markers show baboons that are out-of-sight from the focal location. Each panel corresponds to a different time, as the baboons move across the landscape.

## Step selection analysis

To capture the social and asocial features most predictive of baboon movement decision-making, we carried out a step selection analysis. Step selection functions (related to resource selection functions [*Boyce and McDonald, 1999*]) are models that predict the future location of individual animals based on their current location and heading, and the features of the environment. The models can incorporate both the animal's movement characteristics (typically, its distribution of step lengths and turning angles) and any environmental features that are expected to be relevant, including features of the physical environment such as vegetation density and the locations of roads and rivers, and social features such as the locations and movements of conspecifics. Step selection analyses are increasingly employed to model the habitat preferences and home ranges of wild animals (*Fortin et al., 2005*, *Thurfjell et al., 2014*), but to our knowledge have not before been used to look at within-group social interactions, likely due to the lack of tracking data from multiple animals within a single social group simultaneously. For a review of the application of step selection functions to ecological data, see (*Thurfjell et al., 2014*).

In our analysis, we fit separate step selection models for each individual baboon. To reduce noise, we only fit models for the 16 baboons for which we had complete tracking information for all 14 days, though we used the data from all tracked baboons when incorporating social features into these models.

The process of model fitting and comparison was carried out using the following process, summarized in *Appendix 1—figure 5* (with further details given below):

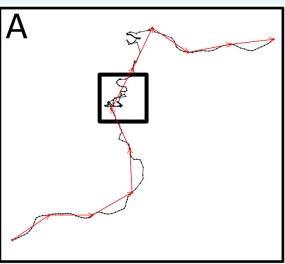

Break individual trajectory into steps.

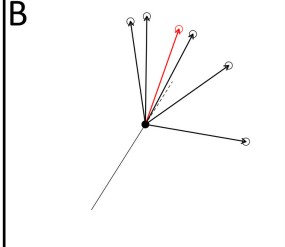

For each step, compare **real (chosen) location** to **alternative locations**…

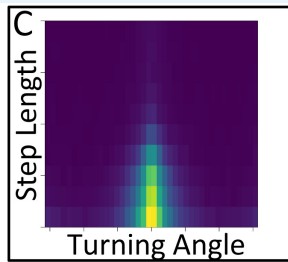

…drawn from individual's step length / turning angle distribution.

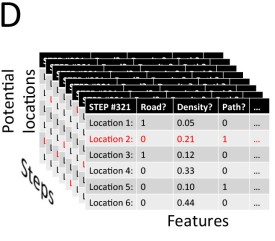

Extract features from each potential location (including **chosen** and **alternative locations**).

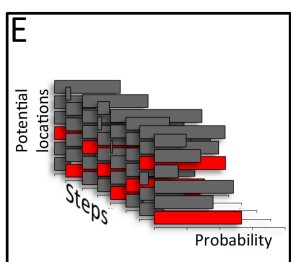

Fit a probabilistic model based on features to distinguish **chosen** location from amongst potential locations for each step.

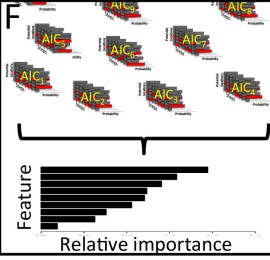

Identify most important features by fitting models containing all possible feature combos and computing weighted AIC scores.

**Appendix 1—figure 5.** Analysis pipeline for determining which features are most important in predicting individual baboons' decisions, using a step selection analysis and multi-model inference. To model a given baboon, its trajectory is first broken up into a series of steps using spatial discretization (**A**). Black line shows the full trajectory, and red lines show the spatially discretized steps. Square encloses one step. (**B**) For each step, the real location chosen by the baboon (red) is then compared to a set of alternative options (black), with these options drawn from that individual's empirical step length / turning angle distribution (**C**). Features (such as whether the location is on a road, the vegetation density, how many baboons are within a certain range, etc.) are then extracted from each potential location, across all steps (**D**). A conditional logistic regression model predicting the probability that the baboon chose each alternative option based on the features of all options (**E**) is then fit using maximum likelihood. (**F**) Models containing all possible combinations of features are fit in this way, and their AIC scores computed. The relative importance of each feature in the models is then determined by computing the AIC weight of each feature across all models. Features that are present in many of the best models receive the largest weights.

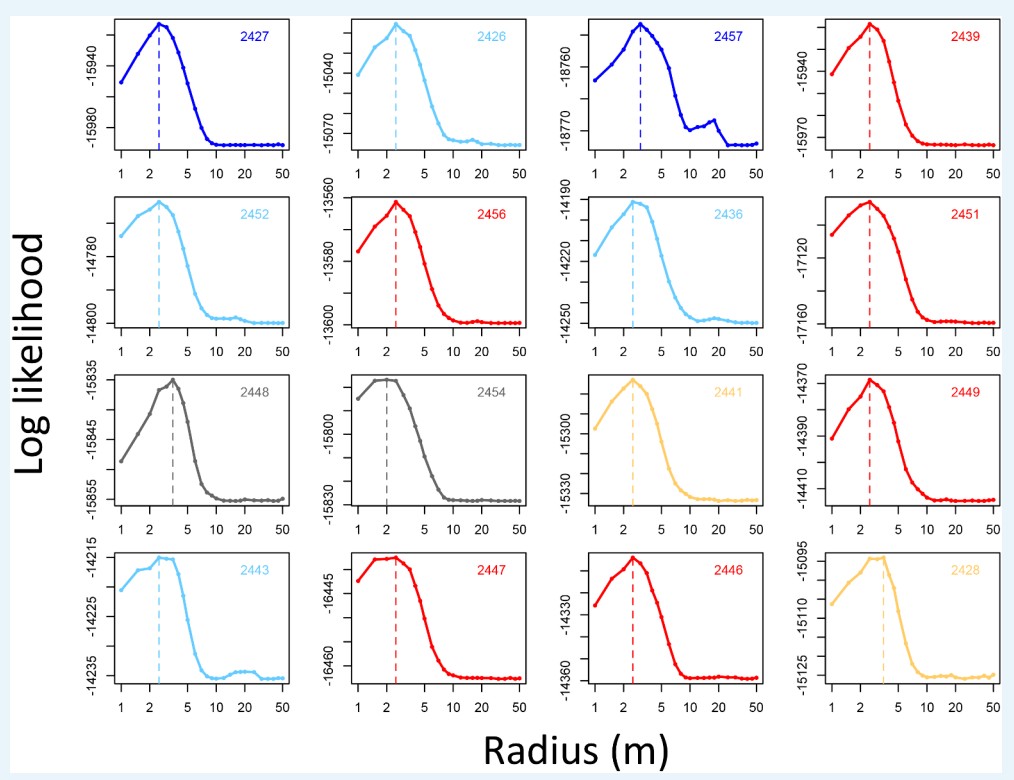

**Appendix 1—figure 6.** Determining the spatial scale over which environment density predicts baboon movement decisions. For each baboon (individual panels), models predicting baboon movement decisions from the environment density (density of non-ground pixels within a radius R) were fit across a range of possible radii. Plots show the log maximum likelihood of models (y-axis) for each value of *R* (x-axis), with the peak value representing the environment density scale best supported by the data.

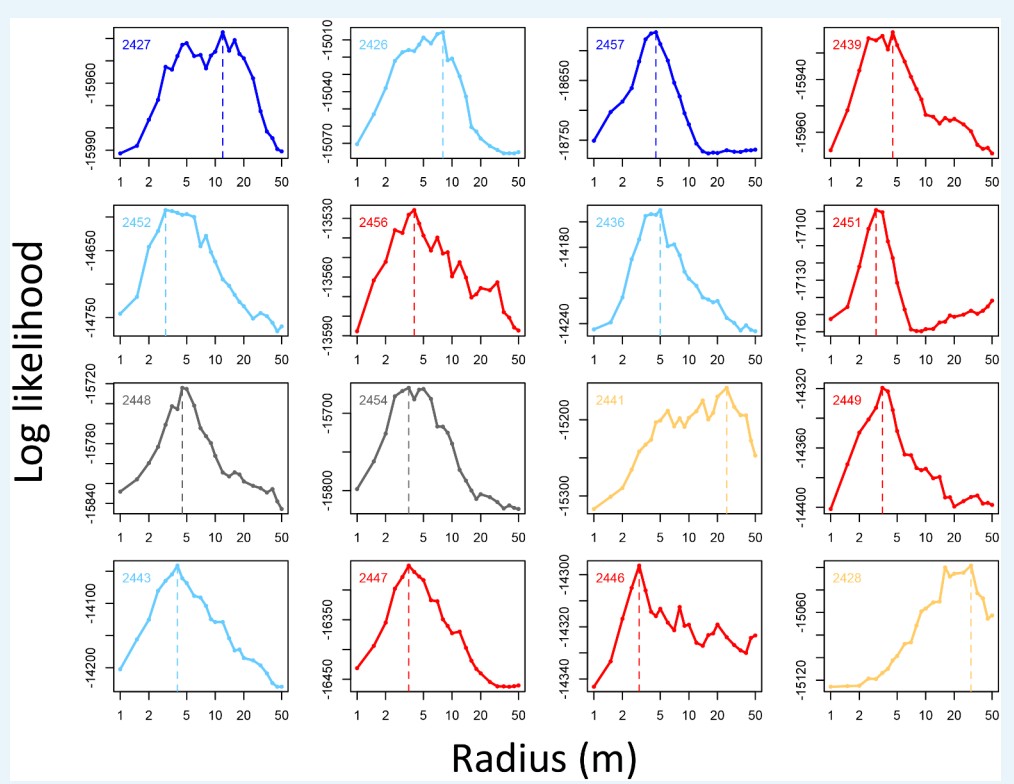

**Appendix 1—figure 7.** Determining the spatial scale over which social density (the fraction of other baboons within a given radius of a potential location) predicts baboon movement decisions. For each baboon (individual panels), models predicting baboon movement decisions from the social density (fraction of other baboons within a radius *R*) were fit across a range of possible radii. Plots show the log maximum likelihood of models (y-axis) for each value of *R* (x-axis), with the peak value representing the social density scale best supported by the data.

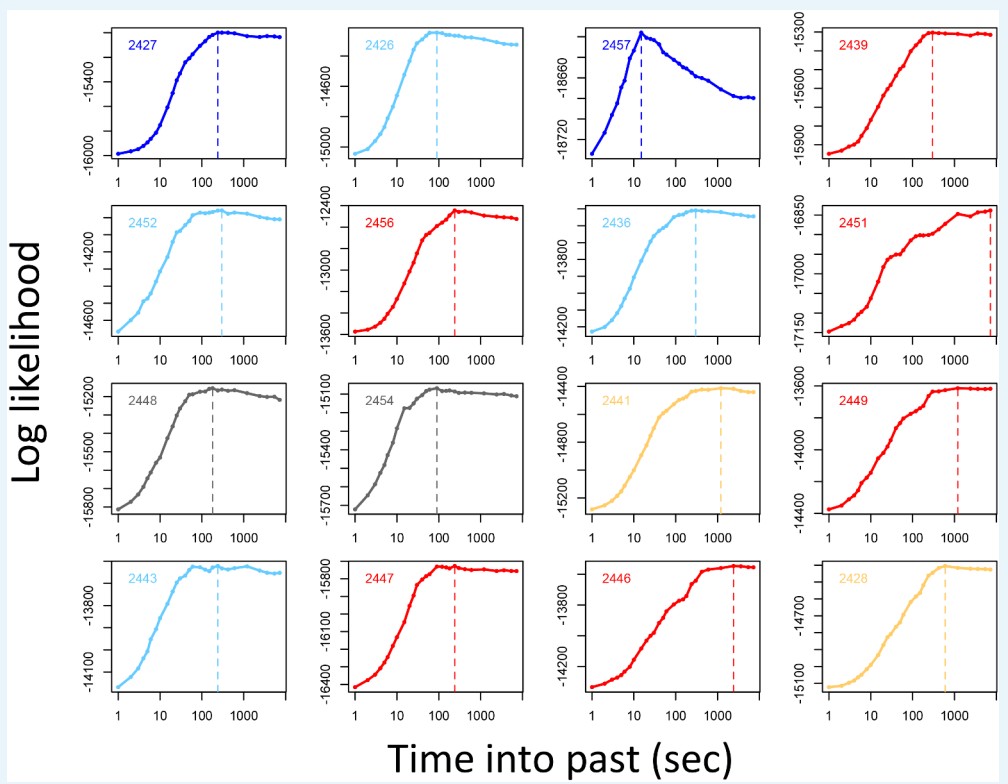

**Appendix 1—figure 8.** Determining the temporal scale over which the previous locations of other baboons predicts baboon movement decisions. For each baboon (individual panels), models of baboon movement decisions were fit using the number of baboons to occupy a potential location within a certain window of time into the past as a predictor variable, across a range of time windows. Plots show the log maximum likelihood of models (y-axis) for each time window (x-axis), with the peak value representing the temporal scale best supported by the data.

1. Spatial discretization: Spatially discretize the individual baboon's trajectory into steps of approximate length $R$. For the analysis presented in the main text, we set $R = 5$ m. Results for an alternate value of $R$ ($R = 10$ m) are shown in **Appendix 1—figure 11**.
2. Determine step length / turning angle distributions: Compute the distribution of step lengths and turning angles over all steps taken.
3. Draw other potential locations: Draw $N$ steps randomly from this distribution. Together with the location associated with the baboon's real step, these represent potential locations that the baboon could have chosen to move. Here, we set $N = 5$.
4. Measure location characteristics: Measure characteristics of the social and asocial environment associated with each potential location. These could include features such as vegetation density at that location, density of other baboons nearby, whether the location is on a road, etc.
5. Fit models: Fit conditional logistic regression models that predict the location that the baboon actually moved (the real location) from all potential location options based on the features used as predictor variables.
6. Assess relative importance of features using multi-model inference: Fit models incorporating all possible combinations of features as predictor variables, then compute the feature importance (AIC weight) of each feature to quantify the relative importance of different features.
7. Assess predictive power: Compare model performance on out-of-sample data to assess the predictive power of different fitted models.

## Step 1. Spatial discretization

Spatial discretization is an alternative to temporal discretization for the analysis of trajectory data. Whereas in temporal discretization, data points are recorded at a constant temporal rate (e.g. one point per second), in spatial discretization points are recorded at a constant spatial rate (e.g. one point for every 5 m moved). Using spatial discretization allows us to focus mainly on where animals moved rather than when they moved. This is important because baboons can spend long periods of time not moving, and a model fit using temporal discretization would be strongly biased by these periods. While the factors that govern a baboon's decision of when to move are interesting, they are not addressed here.

Beginning with a temporally discretized trajectory (1 Hz GPS data), spatial discretization is accomplished as follows. Start at the first location, and record this location as the first in the trajectory. This location becomes the reference location. At each time point, measure the distance between the baboon's current location and the reference location. When this distance exceeds a threshold radius value $R$, the next location is recorded. This location now becomes the new reference location and the process repeats. From this process of spatial discretization, one obtains a trajectory with one point approximately every $R$ meters. An example of spatial discretization of a baboon's path is shown in *Appendix 1—figure 5A*. Note that if temporal sampling were continuous, spatially-discretized steps would be exactly $R$ meters apart. However, in the case of discrete temporal sampling, this distance can be slightly larger than $R$ because sometimes the animal moves from just below a distance $R$ from the reference point, to a distance just above it within a single second.

In the analyses presented here, we used a discretization radius of $R = 5$ m. However, step selection results were qualitatively unchanged when using a radius of 10 m (*Appendix 1—figure 11*).

## Step 2. Step length / turning angle distributions

The distributions of step lengths and turning angles were determined from the spatially-discretized trajectory data for each individual (*Appendix 1—figure 5C*). Due to the nature of spatial discretization, the distribution of step lengths was very narrow, with most step lengths approximately equal to or slightly above $R$.

## Step 3. Draw other potential locations

For each observed step, $N$ step length / turning angle combinations were drawn from the baboon's empirical distribution (determined in Step 2). This yielded $N + 1$ potential locations for each observed step (1 observed step, and $N$ drawn steps; *Appendix 1—figure 5B*). In the analyses presented here, $N = 5$.

## Step 4. Measure location characteristics

We measured various social and nonsocial characteristics associated with each potential location. This yielded a vector of features, which were used in the subsequent model fit (*Appendix 1—figure 5D*). The features used in the model fits are shown in *Table 1*, with further details given in *Appendix 1—table 1*.

**Appendix 1—table 1.** Features (predictor variables) used in conditional logistic regression models to predict baboon movement decisions.

| Feature | Description | Range | Considered as: |
|---|---|---|---|

*Appendix 1—table 1 continued on next page*

*Appendix 1—table 1 continued*

| Feature | Description | Range | Considered as: |
|---|---|---|---|
| Environment density | Fraction of raster pixels containing non-ground points within a specified radius ($R = 2.5$ m) from the potential location (a proxy for vegetation density) | [0, 1] | Habitat Feature |
| Social density | Fraction of baboons currently tracked within a specified radius ($R_{soc} = 4.25$ m) of the potential location | [0, 1] | Social Feature |
| Sleep site direction | The dot product of the direction vector from the current location to the potential location and the vector from the current location to the sleep site location. This quantity ranges from −1 (potential step is in the direct opposite direction from the sleep site) to 1 (potential step is directly toward the sleep site). The influence of the sleep site is expected to vary with time of day, therefore it is fit as an interaction with time of day, yielding three parameters. Times of day included Morning (9 am–12 pm), Midday (12–3 pm), and Evening (3–6 pm) | [−1, 1] | Habitat Feature |
| Roads | A binary predictor indicating whether the potential location is on a road or not. The influence of roads is expected to vary with whether an individual is currently on the road or not, therefore it is fit as an interaction with whether the baboon's current location is on a road (also a binary value), yielding two parameters | {0, 1} | Habitat Feature |
| Recently-used space | An integer indicating how many other baboons (not including the focal individual) have recently (within the past 4.5 min) occupied the potential location, where a baboon is considered to be 'occupying a location' if its position is within 1 m of that location | {0, . . ., N} | Social Feature |
| Ever-used space | A binary variable that is defined as 1 if the location, was ever occupied by another baboon (not the focal baboon) throughout the entire dataset, (past, present, and future) | {0, 1} | Both |
| Animal paths | A binary variable indicating whether a potential location is on an animal path (1) or not (0). It is fit as an interaction with whether the baboon's current location is on a path (also a binary value), yielding two parameters | {0, 1} | Habitat feature |
| Visible neighbors | A continuous variable that indicates the fraction of other group members that are visible from a given location at the time of the step. Group members were defined as visible if a line drawn from the potential location to their location at the relevant time does not pass through any raster cells containing non-ground points. | [0, 1] | Social feature |
| Slope | The change in elevation (in meters) from the starting location to a potential location | $(-\infty, \infty)$ | Habitat feature |

## Step 5. Fit models

Using conditional logistic regression, we fit models that distinguish the actual step taken from the other N steps (drawn from the step length / turning angle distribution), based on the characteristics of each potential location (*Appendix 1—figure 5E*). In these models, the animal's preference, $S_i$ for a given location $i$ is assumed to be a function of a vector of characteristics of that location $\mathbf{x_i} = \{x_{1i}, x_{2i}, x_{ni}\}$ according to the preference function:

$$S_i = f(x_{1i}, x_{2i}, \ldots x_{ni}) = \beta_1 x_{1i} + \beta_2 x_{2i} + \ldots + \beta_n x_{ni} = \sum_{j=1}^{n} \beta_j x_{ji} \qquad (1)$$

The probability of choosing a given location out of all potential locations $S_1 \ldots S_{(N+1)}$ is then given by

$$P(S_i|\{S_1, S_2 \ldots S_{N+1}\}) = \frac{\exp(S_i)}{\exp(S_1) + \exp(S_2) + \ldots + \exp(S_{N+1})} \quad (2)$$

$$= \frac{\exp(\beta_1 x_{1i} + \beta_2 x_{2i} + \ldots + \beta_n x_{ni})}{\exp(\beta_1 x_{11} + \beta_2 x_{21} + \ldots + \beta_n x_{n1}) + \ldots + \exp(\beta_1 x_{1(N+1)} + \beta_2 x_{2(N+1)} + \ldots + \beta_n x_{n(N+1)})} \quad (3)$$

where the denominator is a normalization factor such that the such of probabilities over all potential locations within a given set equals 1. Note that we first checked for multicollinearity of the predictor variables incorporated into each model by computing the variance inflation factor (VIF). Across all predictors (and for each individual), these were found to be less than 5, below the level typically considered problematic (**Chatterjee and Hadi, 2012**).

## Step 6. Assess relative importance of features using multi-model inference

To quantify the relative importance of different social and environmental features, we used multi-model inference. In multi-model inference, rather than fitting a single model, a model is fit for every possible combination of features (predictor variables), and the relative importance of each feature can be assessed based on which features are present in the best models (**Appendix 1—figure 5F**). The feature weight $w_f$ of each feature $f$ (a measure of the feature's relative importance in the models), is defined as

$$w_f = Z^{-1} \sum_i \delta_i^f \exp(AIC_{min} - AIC_i), \quad (4)$$

where the sum runs over all models $i$. $\delta_i^f$ is an indicator variable which equals 1 if feature $f$ is present in model $i$, $AIC_i$ represents the $AIC$ score of model $i$, $AIC_{min}$ is the minimum $AIC$ score over all models, and $Z^{-1}$ is a normalization constant which ensures that the weights of all features sum to 1.

Before performing multi-model inference, data for each individual were first divided into $n = 15$ equally-sized subsets. Relative importance weights were computed for each subset independently, and the median value taken across all $n$ subsets is shown in **Figure 1C**. This process was repeated for different values of $n$ to assess whether results changed depending on the amount of data used in the fits. We found that while the actual values of the weights depended strongly on the number of subsets used, the rank order of features by their relative weight did not change depending on the value of $n$ (**Appendix 1—figure 9**). This suggests that while we cannot claim to quantify the relative importance of different features exactly, we can be fairly confident in their order of importance in predicting baboon movement decisions. We therefore focus our subsequent analysis and interpretation on these rank orders rather than the absolute feature weight values.

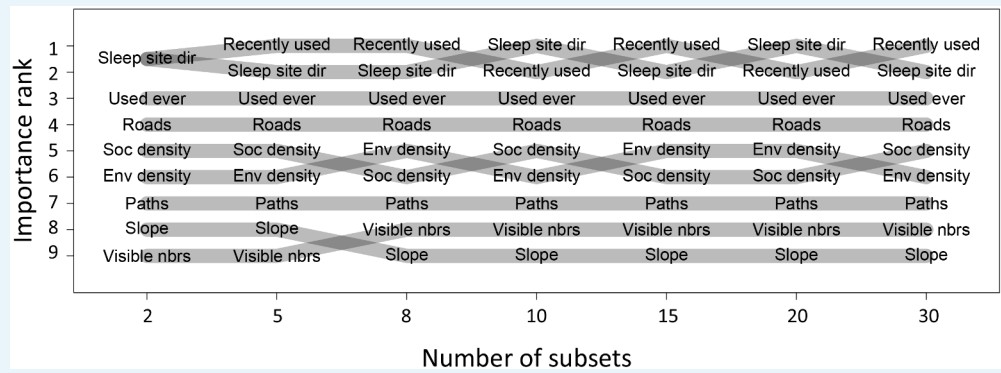

**Appendix 1—figure 9.** Relative importance ranks of habitat and social features do not depend on number of subsets used. Plot shows the relative importance rank (features ranked by AIC weights, y-axis) based on multi-model inference across a range of numbers of subsets used (x-axis). Although the exact values of the AIC weights depend on the number of subsets used, their ranking is minimally changed across a wide range of possible numbers of subsets (few, and only local, swaps between feature ranks). This indicates that the rank order of features is relatively robust to the number of subsets used.

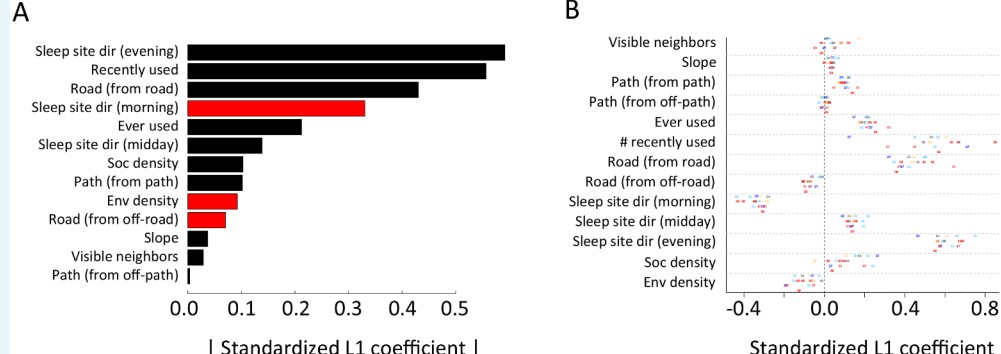

**Appendix 1—figure 10.** Alternative analysis: L1 regularization results. Model coefficients for step selection models fit using L1 regularization as an alternative to multi-model inference. (**A**) Bars represent median L1 standardized coefficients (across all individuals) for each predictor. The length of each bar represents the absolute value of the coefficient, and the color represents the sign of the coefficient (black bars indicate positive coefficients, red bars indicate negative coefficients). (**B**) L1 standardized coefficients for each individual baboon (x-axis) for each predictor (y-axis). Data markers indicate collar numbers for each baboon.

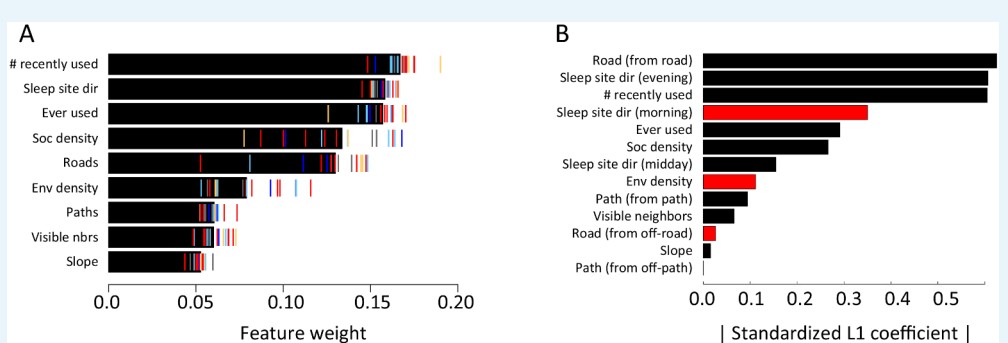

**Appendix 1—figure 11.** Step selection results for an alternative step size of R = 10 m. (**A**) Results of multi-model inference (compare to *Figure 1C*). (**B**) Results of fitting using L1 regularization (compare to *Appendix 1—figure 10*).

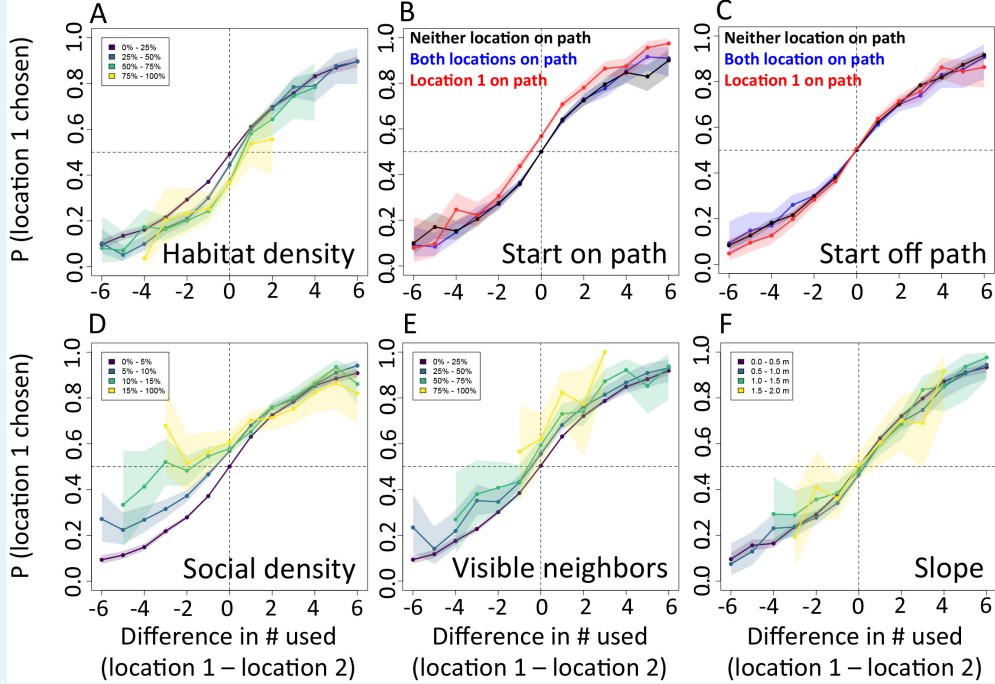

**Appendix 1—figure 12.** The interplay between habitat and social features in shaping individual movement decisions: lower-ranked features. An extension of the analysis shown in *Figure 3*, showing the probability (y-axis) that a baboon chose a given location (location 1) over an alternative location (location 2) as a function the number of baboons to have recently (within the past 4.5 min) occupied each of the two locations (x-axis), under various differences in other habitat and social features. These other features include: (**A**) habitat / vegetation density, (**B–C**) animal paths, (**D**) social density, (**E**) visible neighbors, and (**F**) elevation difference. In panels **A**, **D**, **E**, and **F**, colored lines represent data from various ranges of the difference in the value of the associated feature between the two locations. In panels **B–C**, colored lines represent different possible situations, analogous to those shown in *Figure 3B–C*.

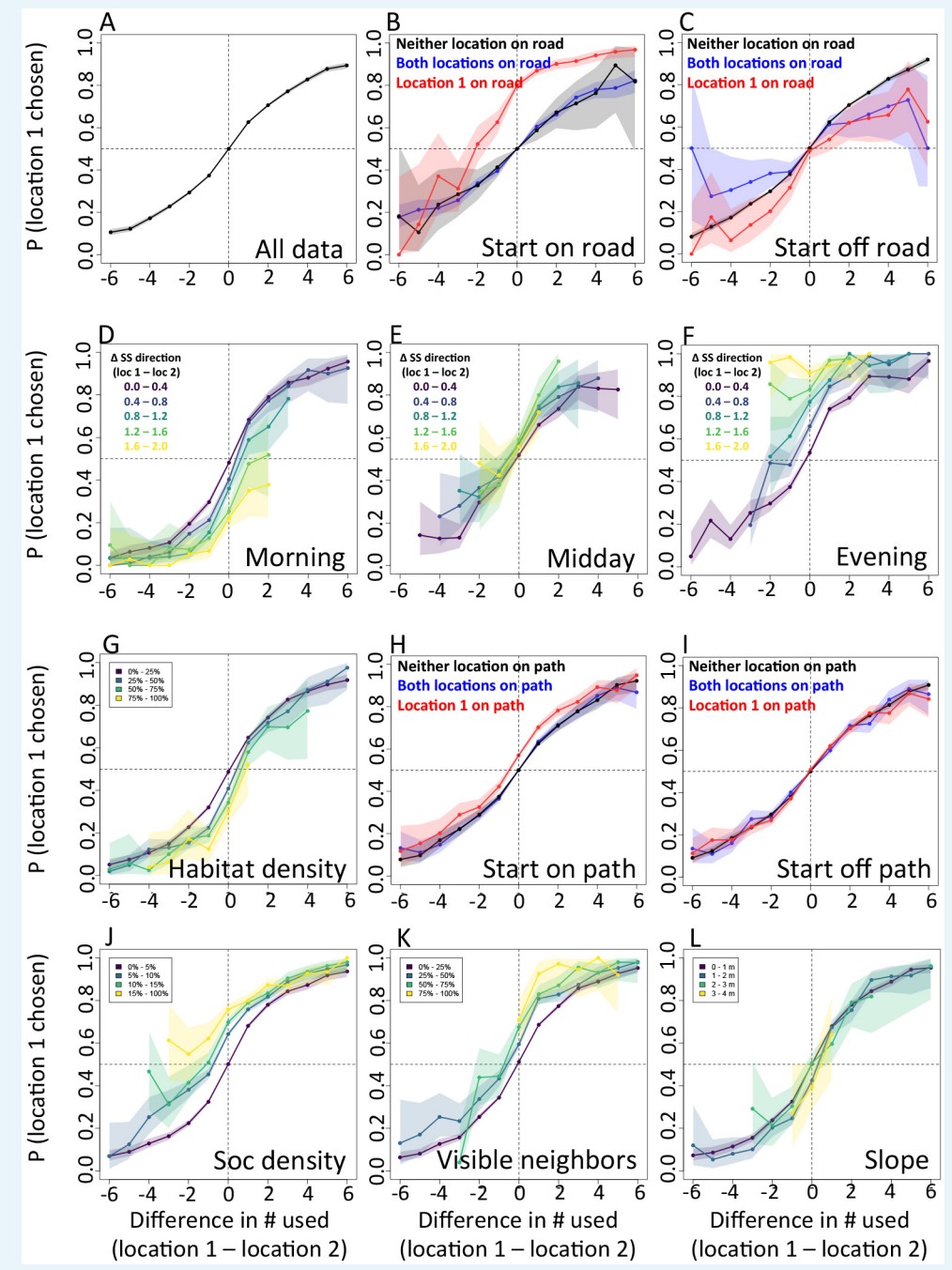

**Appendix 1—figure 13.** The interplay between habitat and social features in shaping individual movement decisions: alternative step size, $R = 10$ m. Analysis is the same as in *Figure 3* (panels **A–F**) and *Appendix 1—figure 12* (panels **G–L**), but using a step size of 10 m instead of the 5 m step size used in the main analysis. Results are qualitatively the same as for the $R = 5$ m case, indicating the robustness of the analysis to the exact step size used.

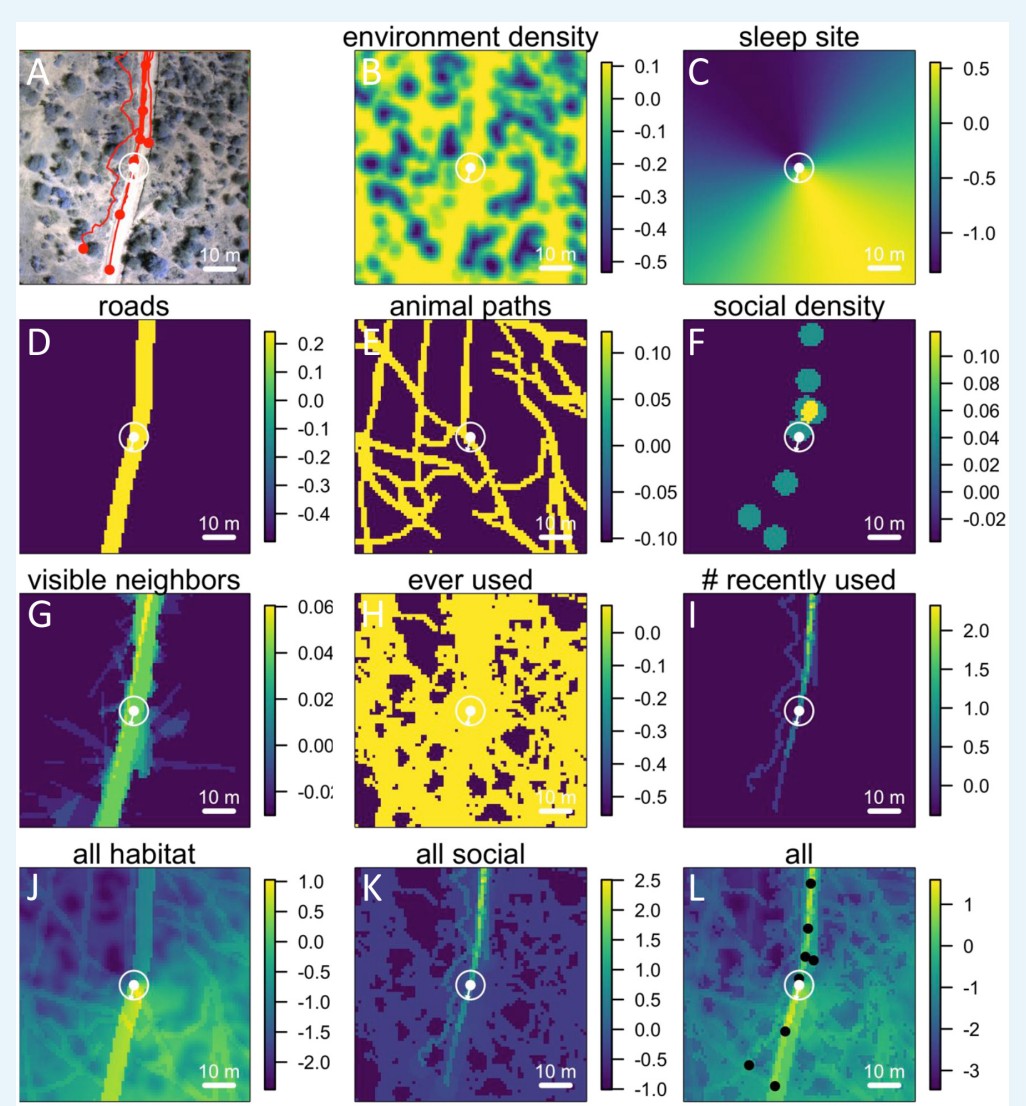

**Appendix 1—figure 14.** Visualizing the preference landscape underlying individual movement decisions: another example. Plots are as in *Figure 2*, but a different example is shown (a case where the focal baboon started on a road).

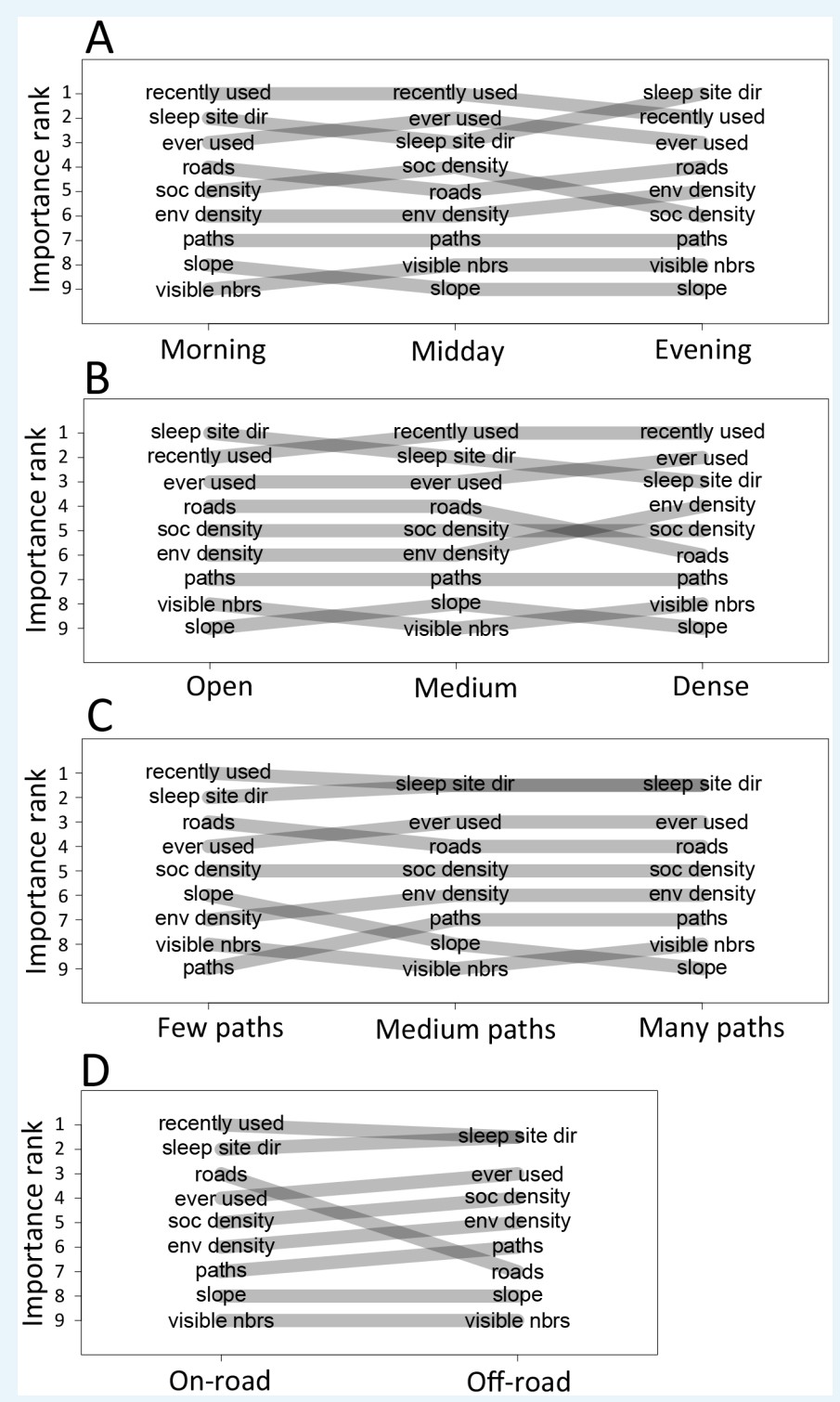

**Appendix 1—figure 15.** The priorities governing individual decisions vary as a function of context. Plots show the importance ranks (ranks of AIC weights) of the different features in step selection models fit using data from each context. (**A**) Models fit for morning, midday, and evening data show that the relative importance rank (ranked AIC weight) of the sleep site direction (sleep site dir) and of roads decreases in the midday, while the relative importance rank of social density (how many other baboons are within a 4.25 m radius of a potential location) increases. (**B**) Models for different habitat densities show that, in

particularly dense environments, the habitat / vegetation density (env density) feature becomes more important, while the sleep site direction (sleep site dir) and roads decrease in relative importance. The environment density of the group is defined as the average density inside the troop's convex hull, and 'open', 'medium', and 'dense' categories represent the bottom, middle, and upper thirds of observed group habitat densities. (**C**) Models for different path densities show that, when the group is in an area with many paths, the relative importance of path-following increases, whereas the importance of ground slope increases when there are fewer paths. Path density was computed as the fraction of the area within the group's convex hull that was located on a path. (**D**) Models for when the group was near a road (convex hull of the group overlapped a road) show high importance of roads, whereas when off a road, roads lose importance in predicting individual decisions.

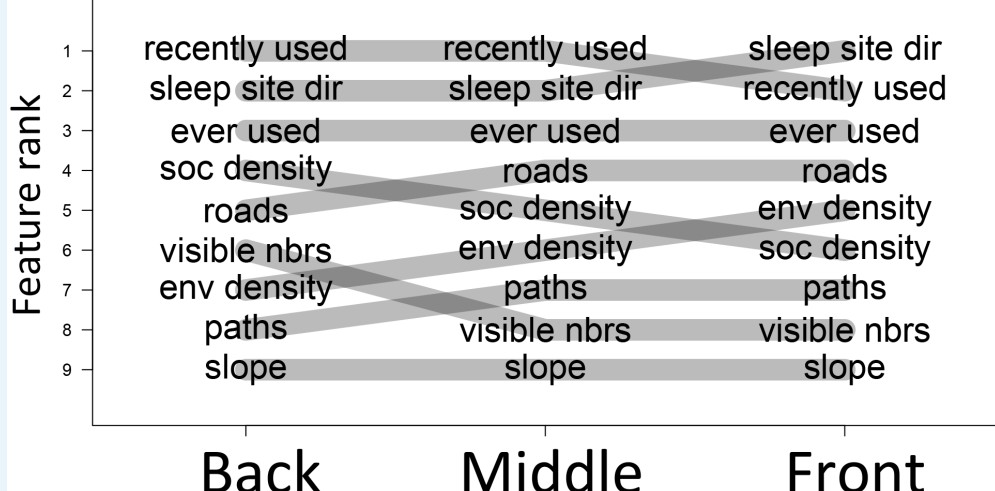

**Appendix 1—figure 16.** The priorities governing individual decisions vary as a function of an individual's current position within the group. In particular, for baboons at the front of the group, habitat features such as environment density (*env density*) and sleep site direction (*sleep site dir*) increase in relative importance, at the expense of social features such as how many baboons are within a 4.25 m radius of a potential location (*soc density*) and how many other baboons had previously occupied a location within the past 4.5 min (*recently used*).

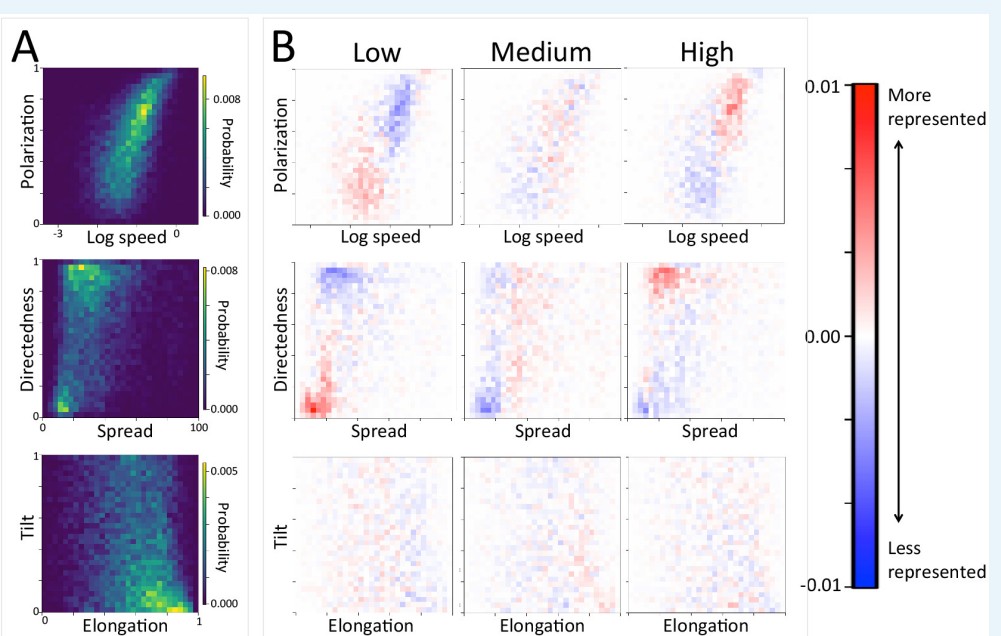

**Appendix 1—figure 17.** The density of paths in an area shapes group-level structure and movement dynamics. (**A**) Two-dimensional histograms of group level properties across all data. Lighter areas indicate group configurations that are more likely to occur in the data. (**B**) Difference between the distributions within a given context (left column: low path density; middle column: medium path density; right column: high path density) and the overall distribution across all data. Redder areas represent group configurations that are over-represented within a given context (relative to the rest of the data), and bluer areas represent under-represented configurations. The path density was defined as the fraction of the area within the convex hull of the group that was on a path. In areas of high path density, the group tends to move faster, and in a more directed fashion. See *Figure 4* and Supplementary methods for descriptions of how group-level properties were computed. See also *Figure 6* and *Appendix 1—figure 21*.

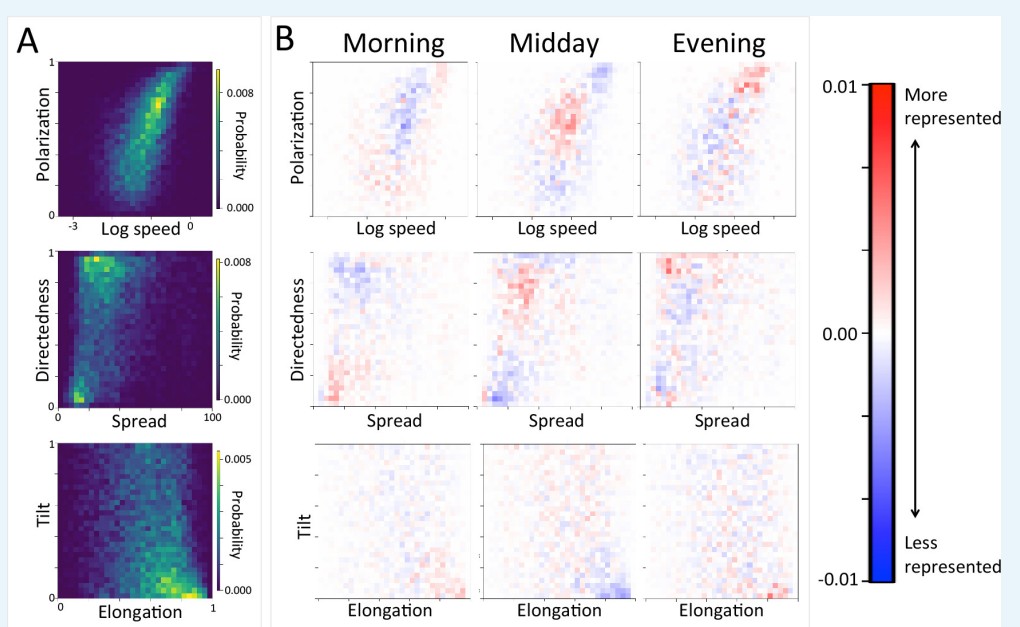

**Appendix 1—figure 18.** Group-level structure and movement dynamics changes as a function of time of day. (**A**) Two-dimensional histograms of group level properties across all data. Lighter areas indicate group configurations that are more likely to occur in the data. (**B**) Difference between the distributions at different times of day (left column: morning; middle column: midday; right column: evening) and the overall distribution across all data. Redder areas represent group configurations that are over-represented within a given context (relative to the rest of the data), and bluer areas represent under-represented configurations. See *Figure 4* and Supplementary methods for descriptions of how group-level properties were computed. See also *Figure 6* and *Appendix 1—figure 22*.

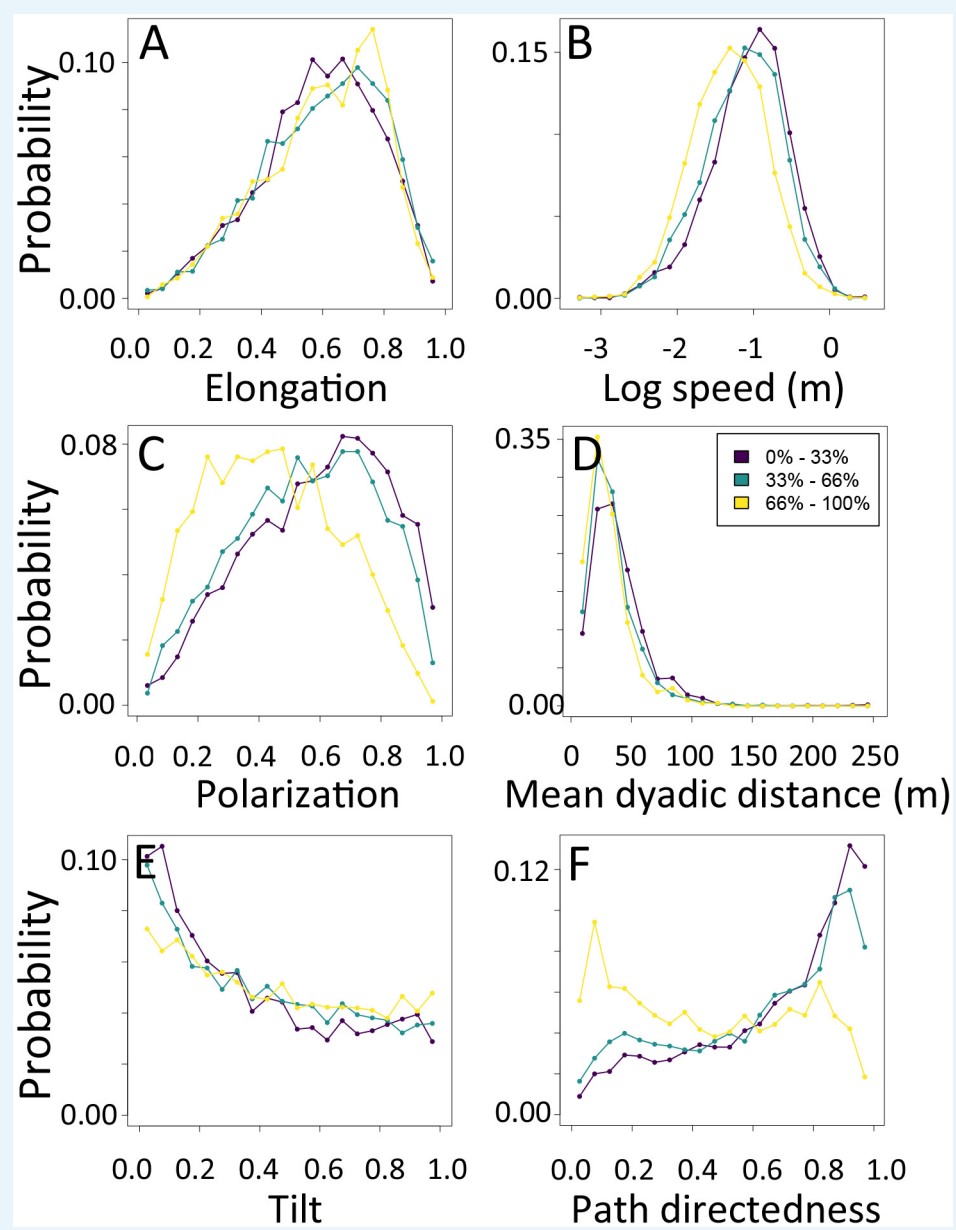

**Appendix 1—figure 19.** 1-dimensional histograms of group-level properties as a function of environment density. Distributions are shown for the highest third of environmental densities experienced by the group (light yellow), the middle third (green), and the bottom third (dark purple).

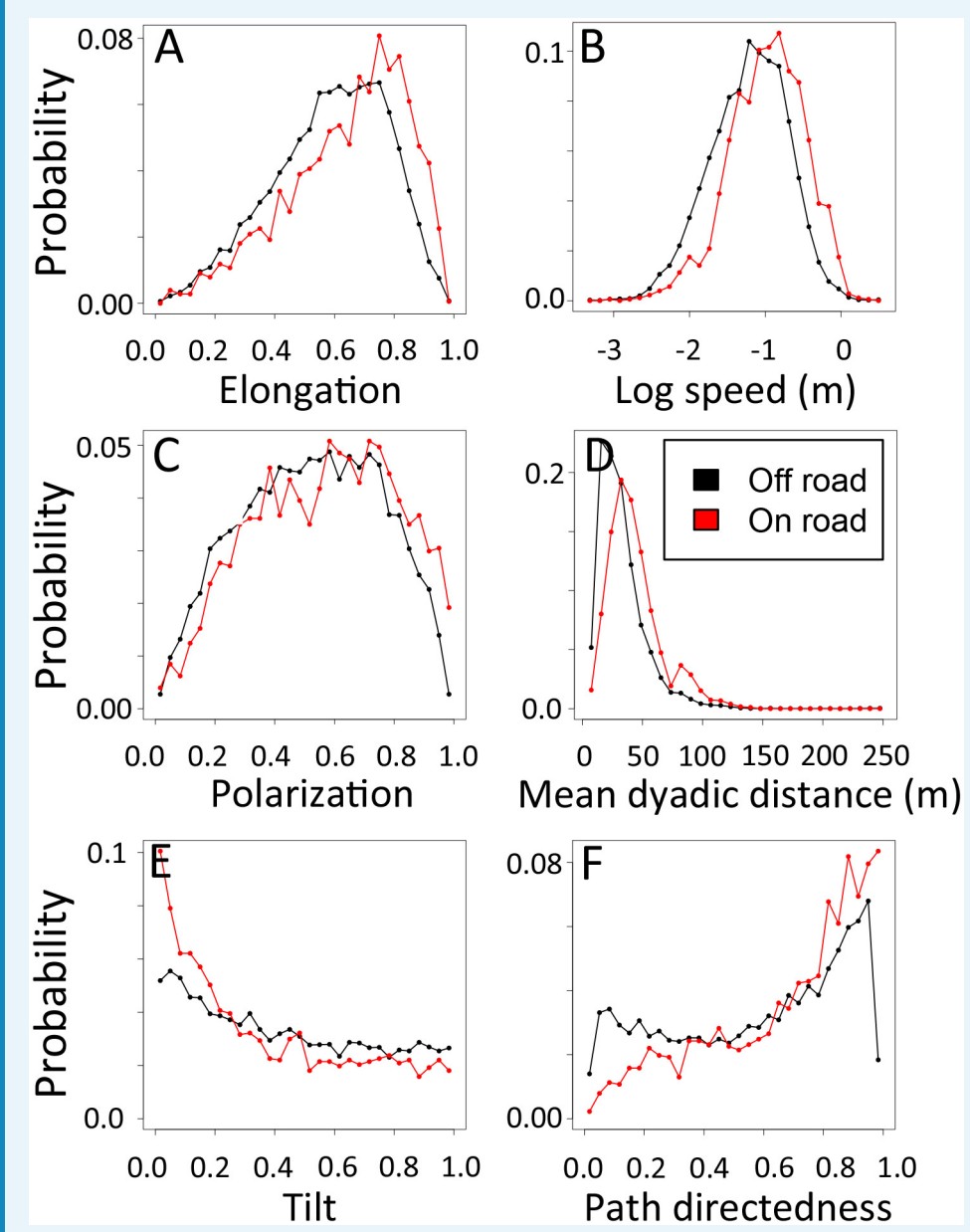

**Appendix 1—figure 20.** 1-dimensional histograms of group-level properties as a function of whether the group was on a road (red) or off-road (black).

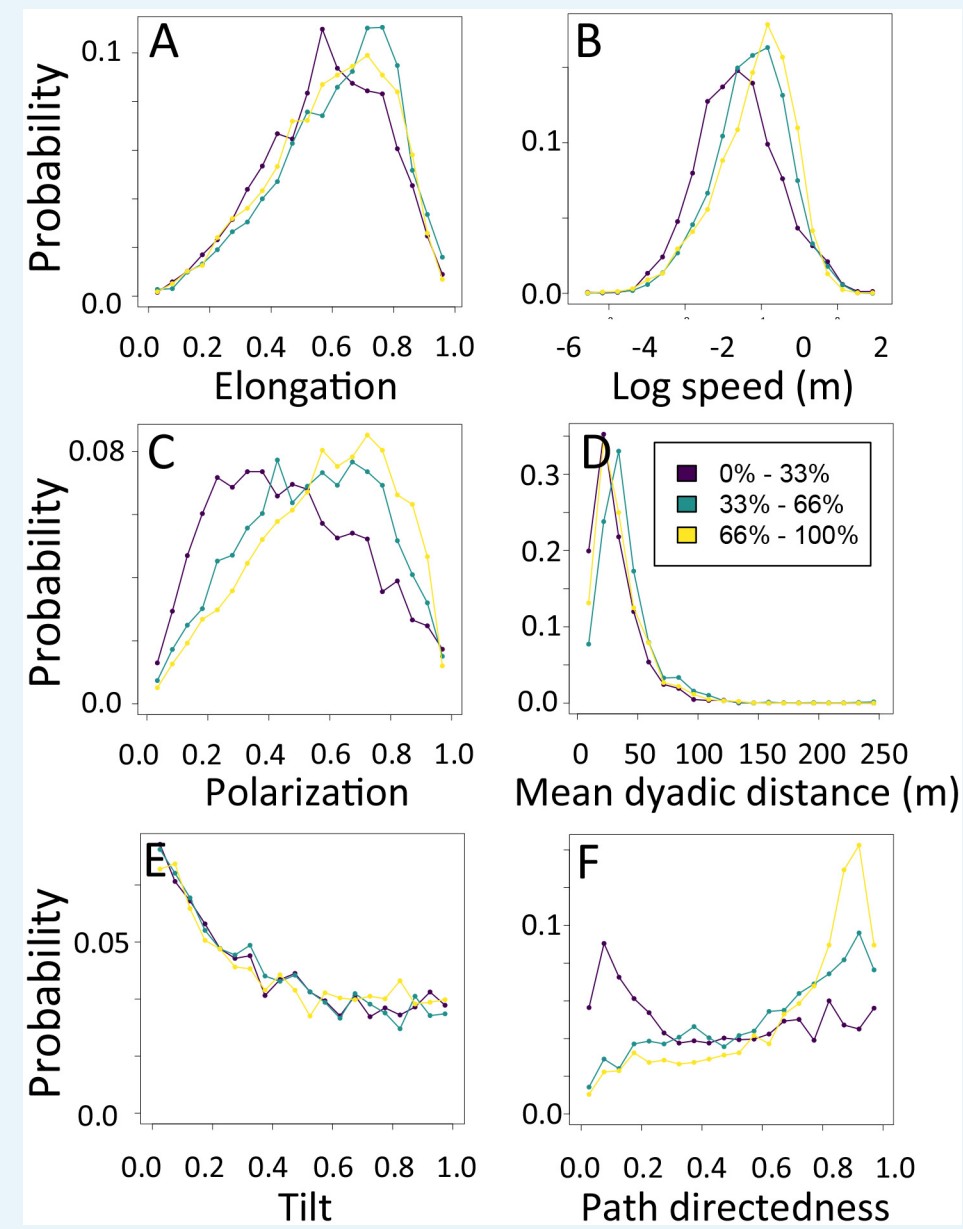

**Appendix 1—figure 21.** 1-dimensional histograms of group-level properties as a function of path density. Distributions are shown for the highest third of path densities experienced by the group (light yellow), the middle third (green), and the bottom third (dark purple).

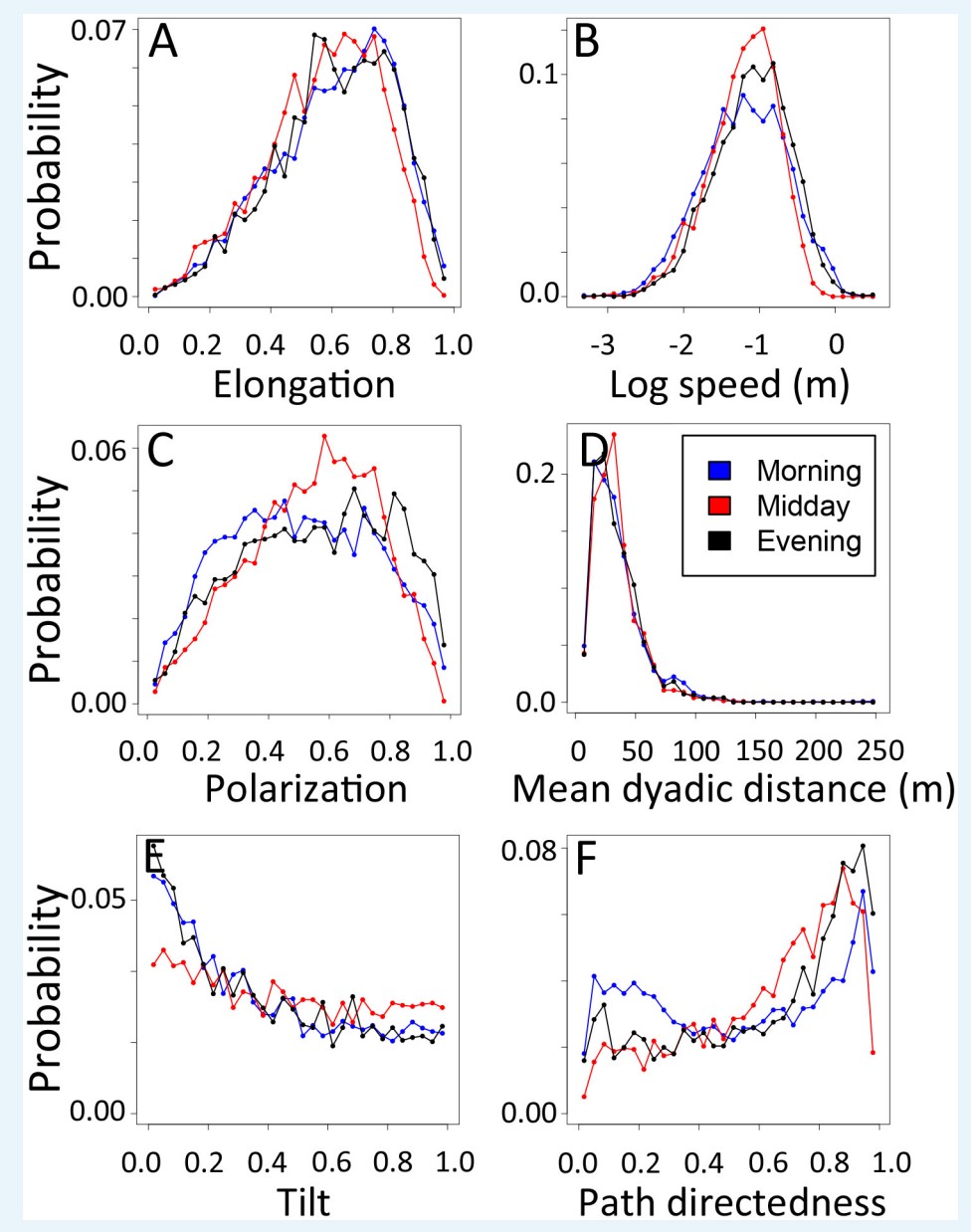

**Appendix 1—figure 22.** 1-dimensional histograms of group-level properties as a function of time of day. Distributions are shown for morning (blue), midday (red), and evening (black).

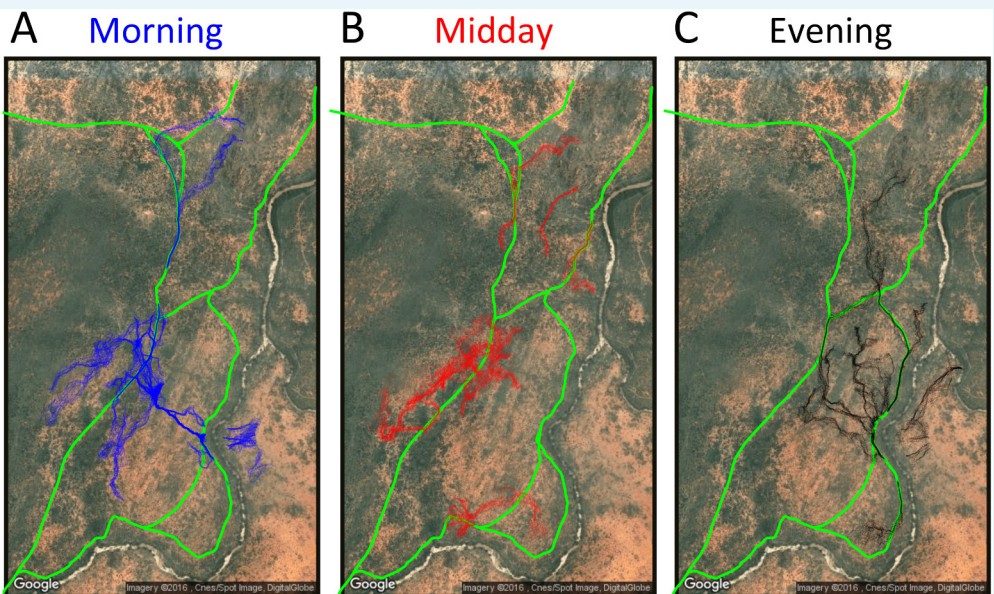

**Appendix 1—figure 23.** Baboon ranging patterns as a function of time of day (A: morning, B: midday, C: evening). Colored points show locations of baboons throughout the first 14 days, and green lines show locations or roads. Baboons spent more time on roads in the morning (12.1%) and evening (12.1%) compared to during the midday period (7.5%).

## Step 7. Assess predictive power

The predictive accuracy of conditional logistic regression models was compared by testing the models' predictive power on data not used in the fitting (out-of-sample data). Before fitting, we divided our dataset into a training dataset (90%) and a test dataset (10%). After fitting the model to the training dataset, we tested its predictive accuracy on the out-of-sample test data. We computed the log loss, a measure of predictive accuracy, defined as

$$L = -\sum_{i=1}^{S} \log P(s_i|M) \tag{5}$$

where $P(s_i|M)$ represents the probability of the observed step $s_i$ given the model $M$ and the sum runs over all data ($S$ observed steps) in the test dataset. This measure can be compared across the fitted models, and can also be compared to a null model which assumes that each of the $N+1$ possible steps is chosen with equal probability, resulting in a null log loss of

$$L_{null} = -S \log\left(\frac{1}{N+1}\right) = -S \log\left(\frac{1}{6}\right) \tag{6}$$

We compared the log loss of four different models: (1) the null model described above, (2) a habitat only model which incorporated only habitat (non-social) features as predictors, (3) a social model which incorporated only social features, and (4) a habitat + social model which incorporated all features. See *Table 1* for a list of the features used in each model, their descriptions, and whether they were considered as habitat features, social features, or both. Note that the *ever used* feature was included in both social only and habitat only models, as well as in the habitat + social model. The reason for including this feature as a predictor in all models was because it likely reflects both social information (where other baboons have been) and habitat information (where it is possible for a baboon to be). This variable is meant to serve as a control for the *recently used* feature, such that any influence

attributed to the *recently used* feature represents the influence of recently-used locations above and beyond the effect of which locations are possible for a baboon to occupy.

Note that in computing the log loss of different models, we used the fits from the L1 regularization analysis (which generated a single model rather than a set of models), described below. We also summed the log loss scores across all baboons to get a overall measure of model predictive abilities.

## Determining relevant spatial and temporal scales

In determining the habitat and social features that are predictive of baboon movement decisions, it is important to select the appropriate spatial scales. For example, baboons could plausibly be responding to environment density over a range of spatial scales, and similarly for social density. We therefore performed a preliminary analysis to select the relevant scales for our subsequent step selection models. Following (*Mashintonio et al., 2014*), we fit a series of models using different smoothing radii and computed the likelihood of each model. We then selected the spatial scale that produced the highest likelihood. This analysis was carried out for both environment/vegetation (*Appendix 1—figure 6*) and social density (*Appendix 1—figure 7*).

We computed the likelihood as a function of the radius for each individual. For each individual, we extracted the radius associated with maximum likelihood, and used the median of these values in subsequent analyses. This optimal radius was found to be 2.5 m for environment density and 4.25 m for social density.

Similarly, when considering locations that have been previously used by other baboons, a question of temporal scale arises. How far into the past should one look for previous use of a location? To answer this question, we performed a similar analysis where the time window over which usage of a location was counted varied from 1 s to 2 hr. Here, we found that the likelihood of the models increased up to approximately 5 min into the past, and then either dropped or stayed relatively flat (*Appendix 1—figure 8*). We therefore selected 4.5 min (the median of the maximum likelihood time windows across all individuals) as our time window for the *recently used* feature.

## Alternative step selection analysis: L1 regularization

To test the robustness of our results to the fitting methodology used, we performed an additional analysis using L1 regularization rather than multi-model inference. L1 regularization is a fitting method that avoids overfitting of models by penalizing models that contain more parameters. In L1 regularization, the best fit model is determined by finding the set of parameters that maximizes the log likelihood minus a penalization term, given by

$$\lambda L_1(\beta) = \lambda \|\beta\| = \lambda \sum_{j=1}^{n} \beta_j \qquad (7)$$

where $L_1(\beta)$ is the L1 norm of the parameter vector $\beta$. The value of $\lambda$ controls how much large parameters are penalized relative to the value of the likelihood - larger values of $\lambda$ favor sparser models. The value of $\lambda$ was chosen using cross-validation, i.e by fitting the model to $4/5$ of the data, computing the prediction error of the left-out $1/5$ of the data, and finding the value of $\lambda$ for which this prediction error was minimized (see [*Reid and Tibshirani, 2014*] for more details). Note that the data used to fit models did not include the 10% of the data which had been previously selected for model validation.

## Analyzing the interplay between *recently used* space and other environmental characteristics

After ranking the measured social and non-social features by their order of importance in predicting baboon movement decisions, we further investigated the influence of three top-ranked features: (1) the number of baboons to have recently occupied a potential location in the past 4.5 min (*recently used* feature), (2) whether a potential location is on- or off-road (*roads* feature), and (3) the direction of a potential location relative to the direction toward the sleep site (*sleep site dir* feature). To elucidate the interplay between these three features, we directly compared their values for the location actually chosen by a baboon with those of a single non-chosen location, drawn from its step length / turning angle distribution. This is equivalent to the way in which step selection models control for the step length / turning angle distribution of individuals, but here rather than fitting models we simply compute the empirical probability of choosing one location (denoted location 1) over the other (denoted location 2) as a function of various features of the two locations.

We first investigated the role of the *recently used* feature by computing this probability as a function of the difference between the number of baboons that have recently occupied location 1 and the number that have recently occupied location 2 (**Figure 3A**). Note that, in this case, either location (the real location or the randomly-drawn location) could be considered as location 1, thus the curve is necessarily symmetric about $x = 0$, and the probability is also by definition equal to 0.5 at $x = 0$.

We next investigated how the incidence of roads on one or both of the locations affected these curves. We considered separately data from when the baboon was initially on a road (**Figure 3B**) and when it was initially off-road (**Figure 3C**). Within each of these data sets, we considered 3 cases: neither of the potential locations was on a road (black line), both locations were on a road (blue line), and one location was on a road whereas the other was not (red line). In this third case, we defined the location on the road to be location 1, and the other to be location 2 (thus allowing us to see how the influence of roads shifted the curve).

We next investigated how a potential location's direction relative to the direction toward the sleep site impacted its probability of being chosen by a baboon (**Figure 3D–F**). Here, we considered separately data from the morning, midday, and evening. For each location, we computed its directedness toward the sleep site (defined as the dot product of the vector pointing to this location from the baboon's starting location with the vector pointing toward the sleep site location from its starting location). For each pair of locations, we defined location 1 as the location with the larger directedness value (i.e. the location closer to the sleep site). We then computed the difference in these values (location 1 – location 2). Finally, we binned these differences into 10 equal-sized bins, ranging from 0 (steps associated with the two locations that are equally directed toward the sleep site) to 2 (steps where location 1 is in a direction directly toward the sleep site, whereas location 2 is in a direction directly away). For all of the data within each bin, we computed the probability of choosing location 1 as a function of the difference between the number of baboons that have occupied location 1 vs. location 2 in the past 4.5 min, as described above.

We also performed similar analysis using the other habitat and social features, including environment density, animal paths, social density, fraction of visible neighbors, and slope. For paths, the procedure was the same as the procedure for roads described above. For the other measures, we computed the difference between the measure's value at location 1 (defined as the location with the larger value) and its value at location 2, then binned these differences into 4 bins (environment density, fraction of visible neighbors: 0–25%, 25–50%, 50–75%, 75–100%; social density: 0–5%, 5–10%, 10–15%, 15–100%; slope: 0–1 m, 1–2 m, 2–3 m, 3–4 m). We then computed probabilities within each bin category as a

function of number of baboons recently using location 1 – number recently using location 2.

To reduce noise, in all plots, data points are only plotted if there were at least 10 instances of a given numerical difference for that context or bin. 95% confidence intervals on probabilities were estimated using Clopper-Pearson intervals (*Agresti, 2002*).

### Definitions of group-level measurements

We quantified the group structure each minute using six measurements. Detailed descriptions, mathematical definitions, theoretical ranges, and units for each measurement are given below.

## Troop speed

*Description:* A measure of how quickly the group as a whole is moving at time $t$. Defined as the distance between consecutive group centroid locations (at time $t$ and time $t + dt$) divided by the time taken to travel that distance ($dt$).

*Mathematical definition:*

$$V(t) = \frac{\sqrt{(X(t+dt) - X(t))^2 + (Y(t+dt) - Y(t))^2}}{dt} \tag{8}$$

where $X(t)$ and $Y(t)$ give the $x$ and $y$ locations of the group centroid (mean position across all individuals in the group) at time $t$.

*Units:* meters / sec.

*Range:* $[0, \infty)$.

## Troop spread (mean dyadic distance)

*Description:* A measure of the overall spread of the troop at time $t$. Computed by measuring the distance between every pair of baboons in the group and taking the mean of these distances.

*Mathematical definition:*

$$S(t) = \frac{1}{N(N-1)} \sum_i^N \sum_{j \neq i} \sqrt{(x_i(t) - x_j(t))^2 + (y_i(t) - y_j(t))^2} \tag{9}$$

where $x_i(t)$ and $y_i(t)$ give the position ($x$ and $y$ coordinates) of individual $i$ at time $t$, and the sums run over all pairs of individuals.

*Units:* meters.

*Range:* $[0, \infty)$.

## Troop polarization

*Description:* A measure of how coordinated individuals are in their direction of travel at time $t$. Computed by adding up the heading vectors of all individuals (normalized to length 1) and dividing by the total number of individuals, then taking the length of the resulting vector. The polarization ranges from 0 to 1, where 0 indicates that headings are pointing in random directions (so that they cancel out), and 1 indicates headings are all pointing in the same direction.

Mathematical definition:

$$P(t) = \frac{1}{N} \sqrt{\left(\sum_{i=1}^{N} h_{i,x}(t)\right)^2 + \left(\sum_{i}^{N} h_{i,y}(t)\right)^2} \tag{10}$$

where $h_{i,x}(t)$ and $h_{i,y)}(t)$ give the $x$ and $y$ components of the heading of individual $i$ at time $t$, and sums run over all $N$ individuals. Note that, to reduce the fluctuations associated with GPS jitter, individual headings are defined as the direction vector pointing to an individual's current position from its position when it was located 5 meters away (i.e. spatially discretized headings).

*Units:* unitless.

*Range:* [0, 1].

## Troop elongation

*Description:* A measure of the aspect ratio of the group at time $t$, ranging from 0 (circular aspect ratio, i.e. the group is as long as it is wide) to 1 (linear structure – the group is much more extended in one direction than the other).

Mathematical Definition:

$$E(t) = 1 - \frac{PC2}{PC1} \tag{11}$$

where $PC1$ and $PC2$ are the lengths of the two principle components derived from a PCA analysis on the positions of all individuals at time $t$. PC2 is defined as the smaller of the principal components.

*Units:* unitless.

*Range:* [0, 1].

## Troop tilt

*Description:* A measure of the group's direction of motion relative to its direction of elongation at time $t$, ranging from 0 (if the group is moving parallel to its long axis, i.e. moving like an arrow) to 1 (if the group is moving perpendicular to its long axis, i.e. moving like a comb).

Mathematical definition:

$$T(t) = \frac{\theta}{\pi/2} \tag{12}$$

where $\theta$ is the angle between the troop's heading (defined as the vector pointing from its position at time t to its position at time $t + dt$) and its long axis (determined using the same PCA analysis as above).

*Units:* unitless.

*Range:* [0, 1].

## Troop path directedness

*Description:* A measure of how directed the troop's movement is in the period from time $t$ to time $t + dt$, equaling 1 if the troop centroid moved directly in a straight line and approaching 0 if the troop centroid moved in a very tortuous path.

Mathematical definition:

$$D(t) = \frac{\delta_i(t \rightarrow t + dt)}{\sqrt{(X(t+dt) - X(t))^2 + (Y(t+dt) - Y(t))^2}} \tag{13}$$

where the numerator $\delta_i(t \rightarrow t + dt)$ is the total path length of the troop centroid and the denominator gives the net displacement of the troop centroid. To reduce the effects of GPS jitter, the centroid trajectory from time $t$ to $t + dt$ is first spatially discretized with a radius $R = 1$ m for each computation.

*Units:* unitless.

*Range:* [0, 1].

