## [Decision Letter]

Thank you for submitting your article "Habitat structure shapes individual decisions and emergent group structure in collectively moving wild baboons" for consideration by *eLife*. Your article has been favorably evaluated by Eve Marder (Senior Editor) and three reviewers, one of whom is a member of our Board of Reviewing Editors. The reviewers have opted to remain anonymous.

The reviewers have discussed the reviews with one another and the Reviewing Editor has drafted this decision to help you prepare a revised submission

The consensus among your reviewers is that your manuscript, on how habitat structure shapes individual decisions and emergent group structure in collectively moving wild baboons, has a well-developed approach to collective behavior, but that you have rather oversold the novelty of your approach. We think that your manuscript would benefit from a more explicit exploration of your results.

One reviewer notes several major concerns. First, there were no specific hypotheses tested in this study, other than the expectation that both social and habitat factors influence individual movement and collective patterns. Both reviewers think it possible that your manuscript could be revised to focus on some hypotheses, for example, about why different features of the habitat are important. In general, *eLife* does not send authors back to collect more data, but a more focused analysis could be beneficial.

Summary:

Your reviewers would like to see a more explicit, hypothesis driven exploration of your results, to explore what features of the habitat influence the group coordination. A second major concern is that your claims of technical advances are not well supported.

Essential revisions:

Technical advances are shared by, for example.

Vanak AT et al. 2013. Moving to stay in place: behavioral mechanisms for coexistence of African large carnivores. Ecology 94, 2619- 2631.

Latombe G et al. 2013. Spatiotemporal dynamics in the response of woodland caribou and moose to the passage of gray wolf. J. Anim. Ecol. 83, 185- 198.

Potts, J. R., Mokross, K., & Lewis, M. A. (2014). A unifying framework for quantifying the nature of animal interactions. Journal of the Royal Society Interface, 11(96), 20140333.

This should be addressed.

At present, the approaches used to link habitat and collective movement are not really compelling. There is concern about your metrics describing collective movement, and how they might be influenced by habitat variables. Since groups can be in different types of habitats, especially at your spatial resolution, you might instead model the collective movement metrics in response to habitat variables. Similarly, it is not clear why you have compared environmental contexts for roads and environmental density but not for other variables.

The leader – follower measure seems largely disconnected from issues of resource availability. Does this imply that baboons are ecologically extreme in some sense such that group cohesion consistently trumps concerns about resource exhaustion and trailing individuals 'losing out' to those they follow through the landscape? Such issues of resource access relative to intra-group position are a fundamental tenet in other studies of the movement of primate groups. Alternatively, are the landscapes simply so lush that there is no risk of resource limitation among group members? Your reviewers would like to see some explanation for this primacy of group cohesion over other considerations.

There is a concern about the short time-duration of their analyzed tracking data. You might include more discussion of why the significant habitat parameters may be biologically important for baboons, more background on the socioecology of olive baboons, and address what could be added with a longer study period.

---

## [Author Response]

*Essential revisions:*

*Technical advances are shared by, for example.*

*Vanak AT et al. 2013. Moving to stay in place: behavioral mechanisms for coexistence of African large carnivores. Ecology 94, 2619- 2631.*

*Latombe G et al. 2013. Spatiotemporal dynamics in the response of woodland caribou and moose to the passage of gray wolf. J. Anim. Ecol. 83, 185- 198.*

*Potts, J. R., Mokross, K., & Lewis, M. A. (2014). A unifying framework for quantifying the nature of animal interactions. Journal of the Royal Society Interface, 11(96), 20140333.*

*This should be addressed.*

We thank the reviewers for pointing us to these references, and for helping focus us on what are actually the main contributions of our work. We have made an effort throughout the manuscript to remove any claims about the novelty of studying habitat and social influences on movement within the same framework, which the reviewers rightly point out has been addressed in recent work (and which we have now referenced where relevant). Instead, we focus more specifically on integrating an exploration of habitat influences on movement into our understanding of collective behavior – both by disentangling within-group social interactions from interactions with fine-scale habitat features within wild animal groups, and by analyzing how the local habitat shapes emergent group-level properties, which we do think are the more novel and interesting aspects of our study.

*At present, the approaches used to link habitat and collective movement are not really compelling. There is concern about your metrics describing collective movement, and how they might be influenced by habitat variables. Since groups can be in different types of habitats, especially at your spatial resolution, you might instead model the collective movement metrics in response to habitat variables.*

New analyses have now been added to address this concern. We now include an analysis based on linear models and multi-model inference where we explicitly model the relationship between the various habitat variables and the group structure parameters to reveal which features best predict group-level properties (see Figure 6). Our results supplement the trends observed from the original histograms in which we presented how group shapes varies with habitat context. The additional results now also clarify the relative importance of these effects. Most importantly, the results reveal that habitat influences on group structure consistently outweigh the influence of time of day. We thank the reviewer for this suggestion, which has led to a very nice addition to our manuscript.

*Similarly, it is not clear why you have compared environmental contexts for roads and environmental density but not for other variables.*

We now include an analysis of how “path density” (the fraction of the area within the convex hull of the group that is covered by paths) impacts group-level properties. We present both the one and two-dimensional histogram analyses for path density, as well as including these in the modeling analysis described above. We find that group speed, spread, directedness, polarization, elongation, and tilt all increase in areas with many paths (see Figure 23 and Figure 27 for the histogram analysis and Figure 6 for the modeling analysis).

*The leader – follower measure seems largely disconnected from issues of resource availability. Does this imply that baboons are ecologically extreme in some sense such that group cohesion consistently trumps concerns about resource exhaustion and trailing individuals 'losing out' to those they follow through the landscape? Such issues of resource access relative to intra-group position are a fundamental tenet in other studies of the movement of primate groups. Alternatively, are the landscapes simply so lush that there is no risk of resource limitation among group members? Your reviewers would like to see some explanation for this primacy of group cohesion over other considerations.*

The reviewers raise an interesting point about the relationship between the “recently used space” predictor and issues of resource availability. Unfortunately, with our current data we cannot directly address the issue of resource distribution, nor how these resources may deplete as baboons move through the landscape. However, we have added a more in-depth discussion of this result (in the Discussion section), where we consider a set of hypotheses as to why this effect comes out as so important. We also suggest future work which could help discriminate among these ideas:

*“*Our analysis also uncovers a primary role of social interactions – in particular a baboon’s tendency to follow at a very fine scale where others have gone before – in shaping individual movement decisions. […] Regardless of the explanation, baboons’ tendency to follow in the footsteps of others is likely to have consequences for the aggregate space use patterns of baboon troops, with troops making use of a smaller portion of the area within their surroundings than would be naively expected if individuals were moving more randomly within the troop’s boundaries, which in turn could alter their impact on the environment around them.”

*There is a concern about the short time-duration of their analyzed tracking data. You might include more discussion of why the significant habitat parameters may be biologically important for baboons, more background on the socioecology of olive baboons, and address what could be added with a longer study period.*

We have now included more background on olive baboons (see Introduction) as well as more biological interpretation of our findings and a discussion of the limitations of our current study and what insights future work, including longer-term monitoring, might bring (see the Discussion section and various other changes throughout).